# Efficient Kerr soliton comb generation in micro-resonator with interferometric back-coupling

J. M. Chavez Boggio [1✉], D. Bodenmüller [1], S. Ahmed [1], S. Wabnitz [2,3], D. Modotto [4] & T. Hansson[5]

Nonlinear Kerr micro-resonators have enabled fundamental breakthroughs in the understanding of dissipative solitons, as well as in their application to optical frequency comb generation. However, the conversion efficiency of the pump power into a soliton frequency comb typically remains below a few percent. We fabricate and characterize a hybrid Mach-Zehnder ring resonator geometry, consisting of a micro-ring resonator embedded in an additional cavity with twice the optical path length of the ring. The resulting interferometric back coupling enables to achieve an unprecedented control of the pump depletion: pump-to-frequency comb conversion efficiencies of up to 55% of the input pump power is experimentally demonstrated with a soliton crystal comb. We assess the robustness of the proposed on-chip geometry by generating a large variety of dissipative Kerr soliton combs, which require a lower amount of pump power to be accessed, when compared with an isolated micro-ring resonator with identical parameters. Micro-resonators with feedback enable accessing new regimes of coherent soliton comb generation, and are well suited for comb applications in astronomy, spectroscopy and telecommunications.

[1] innoFSPEC-Leibniz Institut für Astrophysik Potsdam, An der Sternwarte 16, 14482 Potsdam, Germany. [2] Dipartimento di Ingegneria dell'Informazione, Elettronica e Telecomunicazioni, Sapienza Università di Roma, via Eudossiana 18, 00184 Rome, Italy. [3] CNR-INO, Istituto Nazionale di Ottica, Via Campi Flegrei 34, 80078 Pozzuoli (NA), Italy. [4] Dipartimento di Ingegneria dell'Informazione, Università di Brescia, via Branze 38, 25123 Brescia, Italy. [5] Department of Physics, Chemistry and Biology, Linköping University, SE-581 83 Linköping, Sweden. ✉email: jboggio2006@gmail.com

State-of-the-art complementary metal-oxide semiconductor (CMOS) technology allows for the fabrication of arbitrary architectures with dimension accuracy approaching the nanometer level[1,2]. This has enabled the development of a variety of chip-based photonic devices with advanced functionalities. A relevant example is provided by optical frequency comb (OFC) generators based on integrated ring resonators[3–7]. Fabricated by using dielectric or semiconductor materials, on-chip resonators are expected to downscale the size of OFCs, offering miniature solutions for a number of applications ranging from tele-communications to metrology, astronomy and spectroscopy[8–15]. In its standard configuration, a bus waveguide is positioned in close proximity to a (ring) resonator, in order to couple pump light through its evanescent field[16–24]. Because of its simplicity, this geometry has widespread use, and it has allowed break-throughs in the generation of coherent OFCs based on dissipative Kerr solitons (DKSs)[25–32]. Nonlinear light propagation in Kerr micro-resonators is accurately modeled with the help of the so-called Lugiato–Lefever equation, which has permitted for a deeper understanding of the soliton generation dynamics[33–39].

A key limitation of present DKS-based micro-resonator OFC sources, though, is the extremely low conversion efficiency of the pump power into the power of the comb lines[40]. This is notor-iously the case when generating bright solitons in the anomalous dispersion region of the ring waveguide[25]. The low transfer of pump power into the comb is partially due to a pump/resonator transverse mode mismatch. More importantly, since solitons are generated for an effective red detuning of the pump from micro-cavity resonances, most of the pump power is reflected, hence it is not coupled into the micro-resonator. In addition, the pump-to-comb conversion efficiency is limited because of parametric gain saturation. For a single DKS comb, this results in comb lines carrying less than 1% of the input pump power[7], whereas for multi-solitons or soliton crystals, the comb lines to pump ratio is typically limited to be less than ~5%[30]. Although the generation of dark soliton combs in the normal dispersion regime may bring the pump-to-comb conversion efficiency up to the 30% range[20,41], dark solitons have a narrow spectral bandwidth[41], and exhibit a quite narrow domain of existence in the parameter space (power vs pump-resonance detuning) when compared with bright DKSs.

The development of novel geometry and/or material platforms might enable the generation of frequency combs with much improved performances. In recent years, taking advantage of the high accuracy of CMOS fabrication capabilities, a number of new architectures for generating OFCs have been proposed, mainly by incorporating the coupling of two adjacent resonators[42–48]. For example, it has been shown that, by placing two ring resonators in close proximity, a split of their resonances occurs, leading to the emergence of new DKS dynamics[44]. Interestingly, the use of two adjacent cavities enables nearly 100% pump recycling, where one cavity stores the pump light, while a DKS is generated in the second cavity[45]. Furthermore, a hybrid Mach–Zehnder micro-ring has been recently used to manage the competition between Raman gain and frequency combs[49].

In this work, we fabricate and characterize a different hybrid Mach–Zehnder micro-ring resonator architecture, consisting of a ring where DKSs are generated, embedded in a secondary cavity with twice optical path length, which acts as a feedback section. The interference of the fields from the ring and the feedback section enables unprecedented control of the pump depletion. This permits us to numerically show that more than 80% of the input pump power can be transferred into the comb lines at the device output. The conversion efficiency is only limited by the pump power which is necessary to store in the ring, in order to sustain the DKS propagation. The robustness of the proposed architecture is experimentally demonstrated by generating a large variety of DKSs, with conversion efficiencies up to 55%. Our approach brings Kerr micro-combs from being inherently low-efficiency devices into comb generators possessing efficiencies that even surpass those of laser sources.

## Results

**Ring resonator with optical feedback**. Figure 1a shows the schematic of the proposed device architecture for efficient DKS comb generation. A twisted bus waveguide couples pump light into a ring resonator through its evanescent field at two different coupling locations (marked as 1 and 2). The radius of the ring resonator is $R$, while the nominal distance between the two eva-nescent coupling points is set to $3\pi R$. In this way, two coupled cavities are formed: the first cavity is the ring itself, while the second cavity comprises the twisted bus waveguide and the left semi-circumference of the same ring[50–54]. The power coupling coefficients $\theta_1$ and $\theta_2$ are adjusted by choosing different gaps between the bus waveguide and the ring at nodes 1 and 2, respectively. DKSs are generated in the ring section by injecting a red-detuned continuous wave (CW) pump. Therefore, a fraction

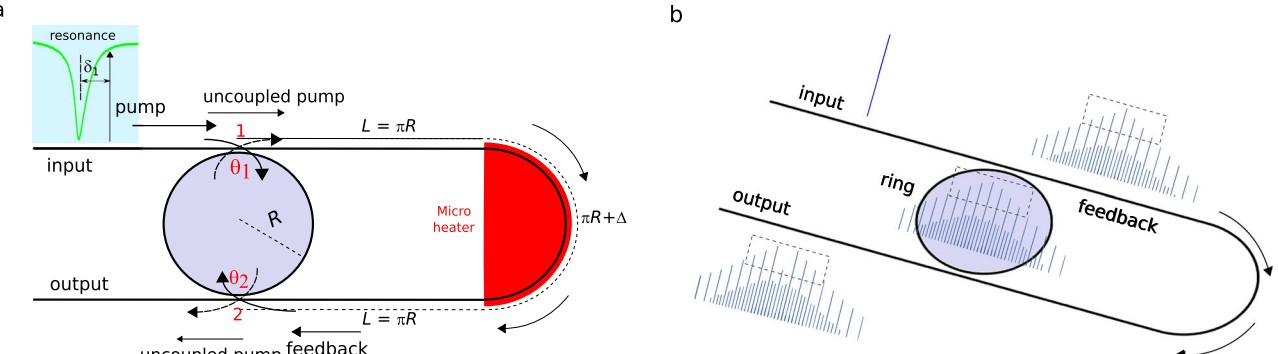

**Fig. 1 Schematic of the ring resonator with an interferometric back-coupling architecture. a** The ring is embedded in a cavity with twice its length. The pump-resonance detuning, $\delta_1$, is in general nonzero. Therefore, at the coupling points 1 and 2, there is a fraction of the field that is in- or out-coupled, while the remaining fraction is reflected. The interference of fields is adjusted by tuning (with a microheater) the length of the feedback section by $\Delta$. Material propagation losses are very low, both in the ring and in the feedback sections, with intrinsic $Q$-factors exceeding $10^6$. **b** Evolution of the frequency comb spectrum at the ring, feedback and output sections. If the pump fields coming out of the ring and feedback sections have similar amplitude and opposite phases, extreme pump depletion occurs at the device output. The amount of circulating pump power in the ring, which is necessary to sustain the solitons, can be controlled by modifying the coupling coefficients.

 

of the pump power is coupled into the ring at node 1, acquiring a phase shift of $\pi/2$, while the remainder of the pump passes through. Furthermore, the DKS field generated in the ring that comes out at node 1 mixes with the pump field that is not coupled into the ring, creating the input field to the feedback section. At node 2, the feedback field and the field coming out from the ring mix, resulting in the device output field. Interestingly, the pump fraction that is in- and then out-coupled (through nodes 1 and 2) acquires a net phase shift of $\pi$. It is possible to obtain complete pump depletion, hence extreme conversion efficiency into the comb lines, if the pump fields coming out from the ring and from the feedback section have: (i) equal amplitudes and (ii) opposite phases. Figure 1b illustrates, in the frequency domain, the principle of the mechanism leading to complete pump depletion. After node 1, the frequency comb spectrum exhibits a strong pump component, as it occurs in an isolated micro-resonator, owing to the uncoupled pump field. On the other hand, inside the ring, the frequency comb spectrum exhibits a pump-to-comb lines ratio that is smaller than in the feedback section. If the intra-cavity power is strong enough, it is possible that the out-coupled pump component at node 2 is equally intense as the pump component from the feedback loop. Whenever these pump fields are out-of-phase, a strong pump depletion occurs at the output, resulting in extremely high conversion efficiency from pump to comb lines. Owing to the conservation of energy/power, the input pump power is distributed among two components: output comb power, and circulating power that is lost in the ring to sustain the solitons.

**Soliton generation with very high conversion efficiency.** The ring resonator with interferometric back-coupling was fabricated by using low-loss silicon nitride ($Si_XN_Y$) with intrinsic $Q$-factors exceeding $10^6$. Microheaters on top of the feedback $Si_XN_Y$ waveguide allow for changing the optical path length of the outer

cavity by a phase amount covering a $2\pi$ period. Figure 2a shows the experimental arrangement and a photography of the chip containing two ring resonators with feedback, used to generate DKSs. Both resonators have an identical radius $R = 800\ \mu m$ and the first gap between bus waveguide and ring (at point 1) 650 nm, whereas the second gap (at point 2) is 410 nm for resonator A and 460 nm for resonator B. The coupling coefficient at node 1 is $\theta_1 \cong 4.1 \times 10^{-3}$ while at node 2 we have $\theta_2 \cong 3.77 \times 10^{-2}$ (for resonator A) or $\theta_2 \cong 2.38 \times 10^{-2}$ (for resonator B), which are obtained through beam propagation simulations (see Methods). Two nominally identical chips were fabricated having the same resonators A and B: in chip 1 wire bonding was added while chip 2 is without wire bonding. The measured dispersion of the free-spectral range (FSR) in the resonator with feedback A is depicted in Fig. 2b. It was obtained by scanning the wavelength of a low-power laser between 1550 and 1630 nm (see Supplementary Note 2).

To generate the solitons, the quasi-TE mode of one resonance of the ring resonator with feedback A was pumped with a CW tunable laser, whose wavelength was tuned from blue to red. We pumped a resonance at 1569.3 nm with a measured loaded $Q$-factor of $0.625 \times 10^6$ with a pump power inside the chip of $P_{in} = 150\ mW$. Figure 2c shows the frequency comb output power from the device, versus pump-resonance detuning, obtained by consecutive scanning of the pump wavelength through the resonance. A Bragg filter was used to completely suppress the pump line. Several steps can be observed in the frequency comb power, which indicates the access to different comb regimes. Since the step heights change between different scans, this indicates that the access to the different comb regimes is done in a stochastic way.

Figure 2e–g shows typical spectra corresponding to the output power traces in Fig. 2c. For Fig. 2h, the pump power was increased to 180 mW. Evenly spaced strong comb lines, accompanied by small comb lines with distinctive spectral

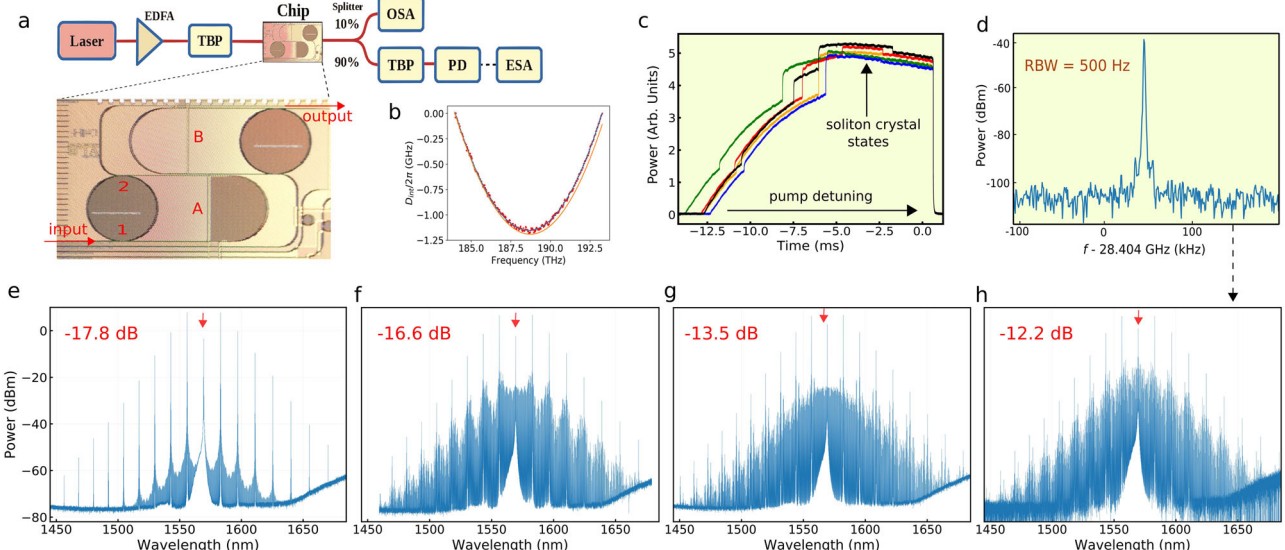

**Fig. 2 Efficient soliton generation in resonators with interferometric back-coupling. a** Experimental setup and picture of a chip containing two ring resonators with feedback (A and B). TBP tunable band-pass filter, PD photodiode, OSA optical spectrum analyzer, ESA electrical spectrum analyzer. The loaded $Q$-factor $= \frac{\omega_0}{\Delta\omega_T}$, where $\omega_0$ is the pump frequency and $\Delta\omega_T$ is the loaded linewidth of the resonance. **b** Measured dispersion of the free-spectral range in resonator A featuring anomalous dispersion. $D_{int}(\mu) = \omega_\mu - (\omega_0 + D_1\mu)$ is the deviation from an ideal equally spaced frequency grid, where $\omega_0$ is the central frequency and $\omega_\mu = \omega_0 + D_1\mu + \frac{1}{2}D_2\mu^2 + \frac{1}{6}D_3\mu^2 + \ldots$ are the frequencies of the resonances. $\frac{D_1}{2\pi}$ is the free-spectral range (FSR) of the micro-resonator while $D_2$ and $D_3$ are the second- and third-order dispersion coefficients. **c** Output power of the generated comb as a function of the pump-resonator detuning. The different colors represent consecutive pump scanning through the resonance. **d** Repetition rate signal of the soliton crystal with defect shown in **h**. Frequency comb spectra featuring a **e** perfect soliton crystal and **f–h** soliton crystals with defect. The pump depletion values indicated in red are measured with respect to the input power.

features, indicate the generation of soliton crystals, as it occurs in isolated Kerr micro-resonators. However, the main difference that can be observed here, with respect to soliton crystals generated in Kerr resonators without feedback, is the strong pump depletion. The spectrum in (e) is a perfect soliton crystal (PSC) exhibiting a 17.8 dB depletion, i.e., only 1.7% of the pump comes out from the micro-resonator with feedback. The pump-to-comb conversion efficiency can be defined as $\epsilon = (P_{\mathrm{comb,out}}/P_{\mathrm{in}})$, where $P_{\mathrm{comb,out}}$ is the output comb power excluding the power at the pump comb line and $P_{\mathrm{in}}$ is the input pump power. By using this definition, the comb lines for the PSC in Fig. 2e carry $\epsilon \sim 55\%$ of the input pump power. The remaining ~45% of the pump power is lost in the ring, in order to sustain the DKSs.

In Fig. 2, and all along this work, soliton crystal states are distinguished from Turing rolls by exhibiting a red-detuned (as opposed to blue-detuned for rolls) pump wavelength: this is confirmed by measuring, with a counter-propagating probe, the pump-resonance wavelength detuning (see Methods). Furthermore, the repetition rate signal was measured for all spectra in Fig. 2f–h, showing the high coherence of soliton crystal with defect generation in resonators with feedback. Figure 2d shows the repetition rate for the spectrum in Fig. 2h.

It is interesting to compare soliton crystal generation in a resonator with feedback, with the soliton crystals obtained in experiments using isolated Kerr micro-resonators with a similar ring radius. In Fig. 1b of[30], a PSC state was generated in a ring with an FSR of 20 GHz, corresponding to 87 solitons, evenly separated by 575 fs. In our case, the PSC in Fig. 2e corresponds to a set of 58 pulses, evenly distributed along the ring circumference with 607 fs spacing. The width of the solitons in Fig. 2e is estimated from a Fourier transform calculation, by using 58 equidistant pulses. A pulse width of 110 fs produces a PSC spectrum that matches well the experimental result. Even though the frequency bandwidth of the PSC in[30] is similar to that in Fig. 2e, conversion efficiencies are drastically different. Due to the absence of the interferometric back-coupling in an isolated micro-ring, the pump is ~17 dB more intense than its immediate comb sidebands. Whereas, in our case, the pump is 11 dB less intense than its nearby comb lines.

In order to model nonlinear pulse propagation in a micro-resonator with feedback, we use an Ikeda map with a delay, associated with the field from the feedback arm. For the Ikeda map simulations we use the parameters of the ring resonator with feedback A that are obtained through measurements and numerical calculations (see Methods). Figure 3a shows the evolution of the frequency spectrum in the micro-ring, while Fig. 3b shows the associated temporal evolution, as the pump-resonance detuning, $\delta_1$, is varied in a resonator with feedback with the same parameters as in Fig. 2, and 180 mW of pump power. The detuning $\delta_1 = -\beta_1 L_1 \Delta\omega$, where $L_1 = \pi R$ and $\beta_1$ is the inverse of the group-velocity (see Supplementary Note 1 for details). The mismatch between the cavities results in a detuning $\delta_2 = 3\delta_1 + \delta_{20}$ with a detuning offset of the feedback arm $\delta_{20} \cong -\beta_0\Delta$, where $\beta_0$ is the propagation constant and $\Delta$ is the length mismatch of the feedback cavity with respect to the nominal double size of the ring cavity. Here the mismatch is set to $\delta_{20} = 1.5$. Note that there is first the formation of Turing rolls at $\delta_1 \sim 0.01$, followed by the merging of those pulses, and the appearance at $\delta_1 \sim 0.027$ of a train of pulses with larger inter-soliton separation. By increasing further the pump detuning, the pulses vanish at $\delta_1 \sim 0.038$. Figure 3c compares the temporal traces at the ring (blue), feedback (green), and output (red) sections, for a detuning of $\delta_1 = 0.033$, where the strongest pump depletion occurs at the device output. The pulses propagating in the ring are sitting on a weak background, while the pulses in the feedback section feature a dark pulse shape. On the other hand,

the output pulses exhibit a strong contrast with the background. We also show the corresponding frequency comb spectra at the three sections: output (Fig. 3e), ring (Fig. 3f), and feedback (Fig. 3g), respectively. Although all spectra look similar, their main difference is found in the intensity of the CW pump component. The pump is around 17 dB stronger than its immediate sidebands in the feedback section, while it is only around 7 dB stronger in the ring section. At the device output, the pump is depleted by as much as 25 dB. For a comparison, Fig. 3d depicts the experimental spectrum, showing an excellent agreement with the simulation. To confirm that the pulses generated at $\delta_1 = 0.033$ indeed correspond to DKSs, we numerically verified that they continue to propagate as single solitons after the removal of all but one pulse (see Supplementary Note 3).

Figure 3h shows the evolution, as a function of detuning $\delta_1$, of the output power of the pump component (blue), comb lines (red), and for the entire comb (dark yellow). There is a range of pump detunings, between $\delta_1 = 0.03$ and 0.035, where the pump is completely depleted, and the conversion efficiency reaches $\epsilon \sim 55\%$. The fact that the total output power drops to 55% of the input power at maximum depletion, is due to the pump power that remains circulating in the ring in order to sustain the DKSs. Interestingly, we may obtain physical insight into the conditions for strong pump depletion to happen, by taking the calculated spectrum in the ring section and multiplying it by the power coupling coefficient $\theta_2 = 3.77 \times 10^{-2}$. In this way, we estimate the field coming out from the ring at node 2, and we find that, for the pump component, it has almost the same intensity as that emerging from the feedback arm. This can be seen in the inset of Fig. 3h, where the three spectra (re-scaled ring, feedback and output) are shifted in wavelength, for better clarity. Note that, contrary to the pump, the comb lines from the (re-scaled) ring and feedback sections have very dissimilar intensities. The corresponding calculated phases for the pump and comb lines are presented in Supplementary Note 3: it turns out that there is a phase difference of $\pi$ for the pump fields, while the other comb lines have smaller phase differences.

**Pump detuning and access to diverse soliton states.** Soliton dynamics in isolated micro-ring resonators are governed by two parameters: the intra-cavity power and the effective detuning[55]. Experimentally, the pump power is kept constant and the detuning is varied to access DKSs: the route that the pump undergoes in the parameter space determines which particular DKS or chaotic state is going to be accessed. A convenient way to compare Kerr resonators having different parameters (such as FSR, nonlinearity, Q-factor, etc.) is obtained by using a normalized pump amplitude $f$, where $f^2 = \frac{8g\eta}{\Delta\omega_{\mathrm{T}}^2} \frac{P_{\mathrm{in}}}{\hbar\omega_0}$, with $P_{\mathrm{in}}$ being the input pump power, $\omega_0$ is the resonance frequency, $\hbar$ is the reduced Planck constant, $\Delta\omega_{\mathrm{T}} = \Delta\omega_0 + \Delta\omega_{\mathrm{ext}}$ is the total cavity linewidth, which is the sum of the intrinsic linewidth $\Delta\omega_0$ and the coupling linewidth $\Delta\omega_{\mathrm{ext}}$. The coupling efficiency is given by $\eta = \Delta\omega_{\mathrm{ext}}/\Delta\omega_{\mathrm{T}}$. The nonlinear gain, which accounts for the Kerr frequency shift per photon, is $g = (\hbar\omega_0^2 c n_2)/(n^2 V_{\mathrm{eff}})$ (see Methods for details).

It has been shown that by scanning the pump wavelength from blue to red, PSCs are deterministically generated in micro-resonators at relatively low pump powers, and up to a normalized pump amplitude of $f \sim 3$[30]. At $f \sim 3$, a spatiotemporal chaos (STC) regime is reached, and a PSC cannot be any longer generated with 100% probability: soliton crystals with defects start to be generated in a stochastic manner. Furthermore, at a normalized pump amplitude of $f \sim 4$ the transient chaos (TC) regime is reached; PSCs and soliton crystals with defect cannot be any longer accessed and only multi-soliton states are accessed by

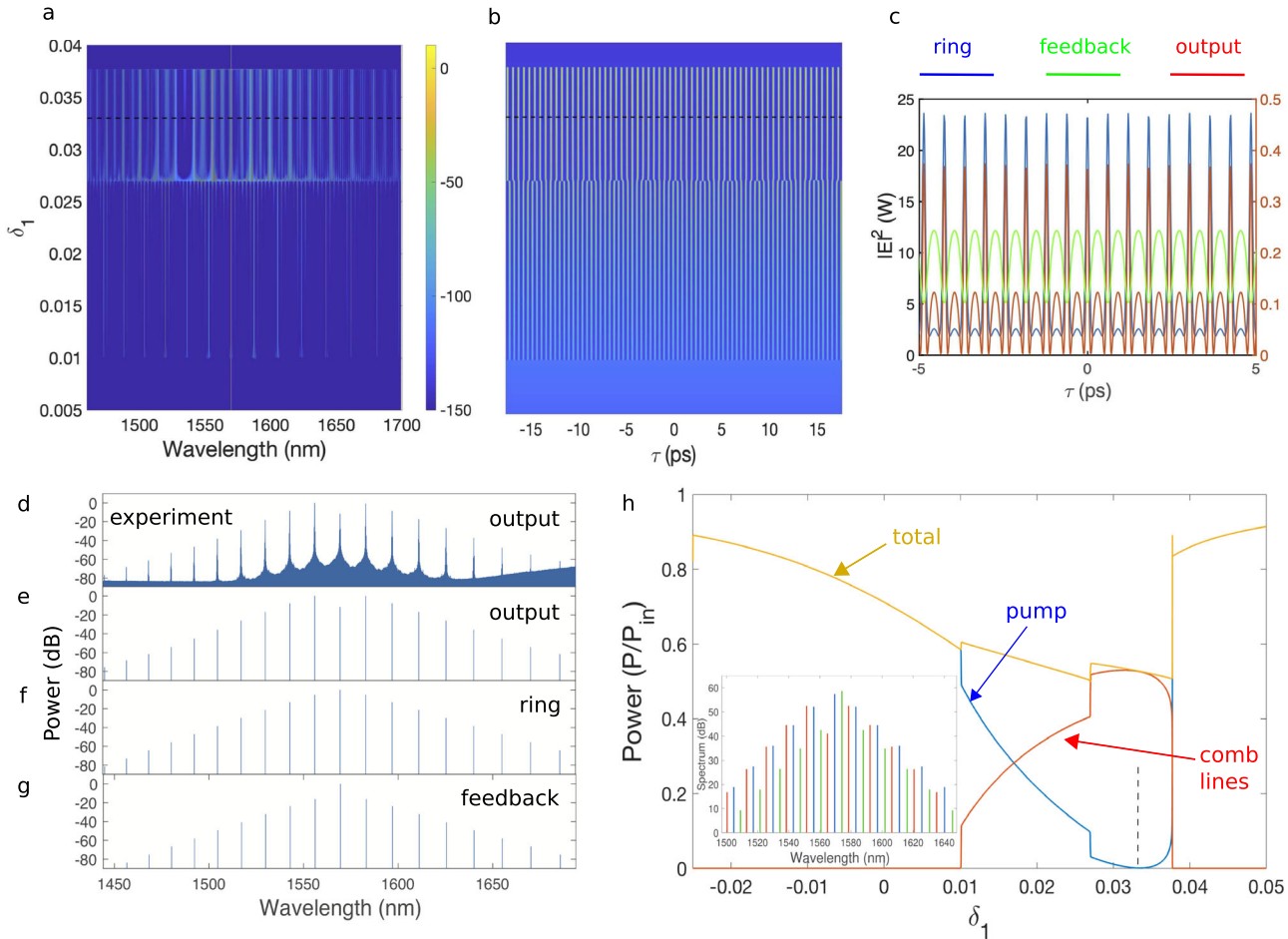

**Fig. 3 Ikeda map simulations.** Evolution as a function of the pump detuning $\delta_1$ of **a** intra-cavity frequency comb spectrum, and **b** intra-cavity temporal intensity. **c** Snapshot of the pulses at $\delta_1 = 0.033$ for ring (blue), feedback (green), and output (red) sections. The left $y$-axis corresponds to the ring power, while the right $y$-axis to the feedback and output powers. Snapshot of the frequency comb spectrum calculated at the **e** output, **f** ring, and **g** feedback sections of the device. **d** The corresponding experimental output spectrum. **h** Calculated power carried by the pump (blue), comb lines (red), and entire comb (dark yellow) vs detuning $\delta_1$. Note that complete pump depletion is obtained for $\delta_1$ between 0.03 and 0.035. The inset of **h** shows the comparison of the comb spectra at the ring (re-scaled, blue), feedback (green), and output (red) sections.

scanning the pump detuning[30]. To compare DKS generation in isolated micro-resonators with that occurring in a resonator with interferometric back-coupling, we show in Fig. 4 traces of the frequency comb output power, versus pump-resonance detuning, obtained for different pump input powers in resonator B. In those measurements, the pump line was removed with a Bragg grating. The relative phase between the cavities was not optimized to enhance the pump depletion and for this case the measured loaded $Q$-factor is $1.8 \times 10^6$ at the pumped resonance. Note that for pump powers of 50 and 68 mW, the traces in Fig. 4 exhibit soliton crystal steps at the highest frequency comb output power values: the steps' slope is positive; hence, it is thermally stable and very easy to access. Successive scannings result in very minor changes in the output power traces, indicating that deterministic access to the PSC states occurs, similar to the case of isolated Kerr resonators. On the other hand, for larger pump powers (i.e., for 80 and 140 mW), different scannings result in frequency comb power traces with soliton steps of different heights, which indicates that the STC regime has been reached. For the 80 mW case for example (Fig. 4c), two types of steps can be noticed: (i) soliton crystal steps that occur with a small power drop and (ii) multi-soliton steps that exhibit a much larger power drop. The upper plot of Fig. 4c shows the spectrum of a soliton crystal with defect, that is generated with 80 mW of pump power. For a pump

power of 140 mW, as it is shown in Fig. 4d, no soliton crystal steps are observed, and only multi-soliton steps are present, which indicates that the TC regime has been reached. Since different step levels indicate the presence of different soliton numbers, from the results in Fig. 4 we may conclude that a large variety of DKS states can be accessed in the resonator-with-feedback architecture.

Interestingly, soliton crystals with defect were even obtained for an input pump power value of $P_{in} \sim 70$ mW, i.e., with $f \sim 1.5$. This corresponds to a pump power four times smaller than the value for an isolated micro-resonator (see Methods for a more detailed description). Errors in determining $f$ might come from an imperfect knowledge of the nonlinear refractive index, and mainly from not knowing the exact pump power that propagates inside the bus waveguide (1 dB error). Even taking into account those error sources, the STC regime starts at $f \sim 2$. Furthermore, the TC regime has its lower boundary at $f \sim 2.8$ instead of $f \sim 4$. Therefore, in Kerr resonators with feedback, some DKS states can be accessed by tuning the pump wavelength from blue to red by using a lower amount of pump power than in an isolated Kerr resonator. This can be related to the pump power re-use, which in our case occurs because of the feedback. Whereas in the case of an isolated micro-resonator the pump that is not coupled into the ring does not contribute to the parametric gain process.

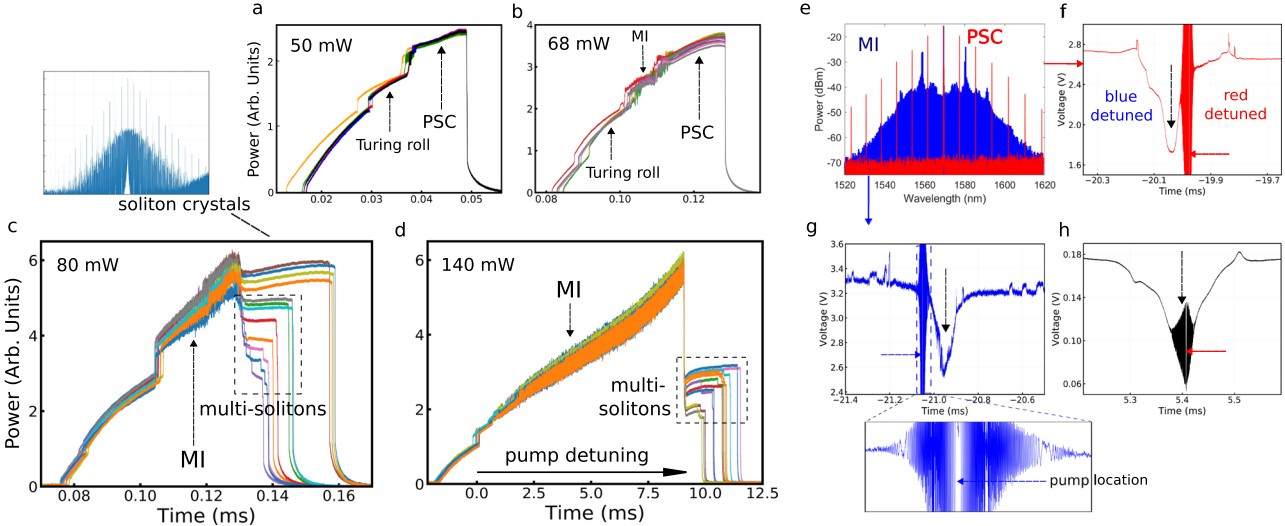

**Fig. 4 Output power of the generated comb as a function of pump-resonance detuning.** The input pump power is **a** 50 mW, **b** 68 mW, **c** 80 mW, and **d** 140 mW. Perfect soliton crystals steps are observed for $P_{in} = 50$, 68, and 80 mW. Multi-soliton steps are observed at 80 and 140 mW. For 80 and 140 mW pump powers, the step height changes for different scannings. The soliton crystal with defect spectrum shown above **c** exhibits a typical spectrum generated at 80 mW, for the detuning that is indicated with a black arrow. **e** MI (blue line) and PSC (red line) spectra. Measurement of the effective pump detuning for the **f** PSC and **g** MI combs. The black arrow in **f**–**h** indicates the center of the resonance. The panel below **g** shows the details of the beating between the pump and the probe. The case in **h** shows the smallest detuning that we could measure in a PSC.

In isolated micro-resonators, DKSs are generated for an effective red-shifted detuning of the pump. For the case of single solitons, the detuning can be many times larger than the resonance linewidth, which results in very poor conversion efficiency from the pump into comb lines. We investigated the effective pump detuning while generating frequency combs by injecting a weak counter-propagating probe, whose wavelength scans the resonance at a 10 Hz rate[21]. The power of the probe is adjusted, in order to match the power of the pump that is back-reflected, owing to imperfections in the resonator. Light from the probe and the reflected pump is filtered with a band-pass filter, in order to reject the comb lines, and it is detected with a fast photodiode. In this way, not only the resonance trace is obtained, but the interference pattern that is produced by the beat between the moving probe and the fixed pump is also recorded. The exact location of the pump corresponds to the position where the beat note frequency vanishes. Figure 4e shows in a red color the spectrum of a PSC, and in blue we illustrate the case of a chaotic modulation instability (MI) state. The spectra correspond to the same resonator and power as in the case of Fig. 4c. Figure 4f, g shows the measurement of the effective detuning for the PSC and MI spectra, respectively. Similar to the case of a single micro-resonator, bistability results in the distinction between chaotic states, which are accessed with a blue-detuned pump wavelength, and soliton states, which are obtained with a red-detuned pump. The panel below Fig. 4g shows the detail of the interference between the pump and the probe lasers. The pump location with respect to the resonance can be accurately measured by retrieving the zero of the beat note frequency.

Soliton crystals (perfect and with defect) were generated with different levels of conversion efficiency, or pump depletion, and their corresponding pump detuning was measured. For all cases, the red detuning was not bigger than ~1.5 times the resonance linewidth. However, no clear correlation between the value of detuning and the associated level of pump depletion was observed, even though we would expect that a smaller detuning could be associated with stronger pump depletion. Figure 4h shows an example of the detuning measurement of a soliton

crystal for which the red detuning is the smallest we have observed.

The complexity of adding a feedback arm to the micro-ring resonator does not seem to drive the device into a generator of chaotic states, but rather into a generator of stable DKSs. Figure 5a–g, i, j shows the variety of coherent and red-detuned frequency comb spectra that can be generated in the ring resonator with feedback. Figure 5a, b shows frequency comb spectra obtained with the chip with wire bonding, where Fig. 5a is obtained with resonator A having Q-factor of $1.23 \times 10^6$, and pumped with $P_{in} = 60$ mW, while Fig. 5b is obtained with resonator B having $Q = 0.54 \times 10^6$, and $P_{in} = 150$ mW. Figure 5c comes from the same set of measurements of Fig. 2, but with 210 mW input pump power. Figure 5d–g, i, j comes from the same set of measurements in Fig. 4 with input powers ranging from 70 to 100 mW. The interference between strong and weak comb lines results in a set of DKSs that are distributed along the resonator circumference, where some pulses are expected to be missing with respect to the perfect SC. The case of Fig. 5h shows an OFC exhibiting the largest pump depletion in our experiments, 22 dB with respect to the input power. Interestingly, it has a repetition rate signal, depicted in (l), which shows a multi-prolonged profile due to fluctuations over a bandwidth of 200 kHz happening at a 1 ms period, that might be associated with a breathing state.

Ikeda map simulations were also performed, in order to retrieve soliton crystals with defects in resonators with interferometric feedback similar to the experimental ones. For the simulation parameters, we used the nominal parameters of resonator B. As input fields, we launched equally spaced pulse trains with some pulses missing. Figure 5i, j (top and bottom) shows the optical fields for two different soliton crystals with defects. The top panels show the temporal profiles in the ring (blue), feedback (green), and output (red), while the middle panels show the corresponding soliton crystal spectra. By comparing the simulations with the experimentally obtained spectra (bottom panels), it can be noticed a very good agreement, which provides verification for the model and an indication of the robustness of soliton propagation in this architecture.

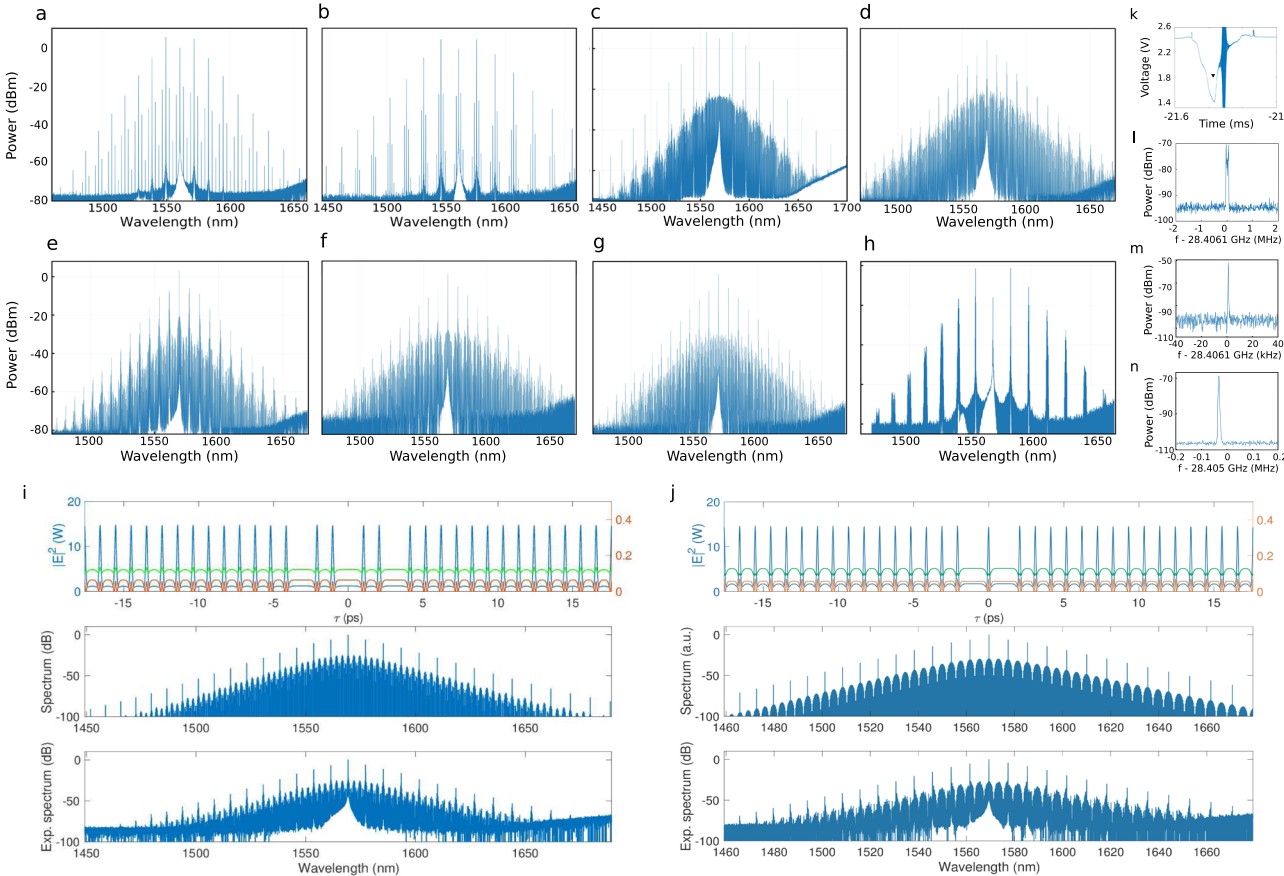

**Fig. 5 Frequency comb spectra with distinctive patterns of intensity variations in the comb lines. a, b** OFC generated in the chip with wire bonding, featuring a transition spectrum between PSC and MI. **c–g, i, j** Soliton crystals with defect obtained with resonator A (**c**) and resonator B (**d–g, i, j**). The spectrum in **h** was obtained for a pump power of 300 mW from the same set of measurements as in Fig. 2, and exhibits a strong enhancement of the comb lines. **k** The effective pump detuning measurement corresponding to the spectrum in **a. l–n** Repetition rate signals of **h, f, g**, respectively. **i, j** Ikeda map simulations of soliton crystal with defects, where the temporal profiles for the ring (blue), feedback (green), and output (red) are shown in the upper panel. The simulated (middle panel) and experimental (bottom panel) comb spectra exhibit a very good agreement.

Although an infinite range of defects and shifts in the soliton locations are possible, results that have been reported over the last couple of years by different groups show that some soliton crystals are more likely to be generated than others. For example, the OFCs in Fig. 5a, b, f, g, i were previously reported by using different architectures and materials[26,29,30,32]. The advantage of using a micro-resonator with the extended cavity architecture comes from the much higher conversion efficiency that is available at its output, when compared with the case of a single micro-resonator, as seen in Fig. 5a–c. Figure 5k shows the pump detuning measurement of the spectrum in Fig. 5a, where a red pump detuning can be observed. Figure 5m, n shows the repetition rate signals, measured with a resolution bandwidth of 1 kHz, for the spectra in Figure 5f, g, respectively.

**Tuning of the conversion efficiency**. The strong pump depletion that is obtained with our microcomb architecture originates from the interference at node 2 of the pump fields coming out from the ring and from the feedback section. The adjustment of the interference process relies on tuning of the phase mismatch between the cavities. However, due to the complexity of the feedback architecture, the interplay of the cavity length mismatch and the associated detuning offset, $\delta_{20}$, with the various resonator parameters, such as the detuning $\delta_1$ and the coupling coefficients, might impact the interference process. To investigate this, we retrieve first the phase mismatch, $\delta_{20}$, by measuring the linewidth

of the resonances, since this linewidth depends on the feedback length, $\Delta \cong -\delta_{20}/\beta_0$, which is tuned with the microheater (see Supplementary Note 1)[50,52–54]. The resonances linewidth in our device is approximately given by

$$\Delta\omega_T \cong [\alpha L_1 + (\theta_1 + \theta_2)/2 + \sqrt{\theta_1\theta_2}\cos(\delta_{20})]/\beta_1 L_1, \quad (1)$$

where $\alpha$ is the absorption loss, $L_1 = \pi R$, $\beta_0 = 2\pi n_{\text{eff}}/\lambda$ is the propagation constant, and $\beta_1 = t_R/2L_1$ is the inverse of the group-velocity, with $t_R$ being the round-trip time. Experimentally, we change $\Delta$ up to a corresponding phase shift slightly larger than $2\pi$, by delivering an electrical power into the microheaters. The linewidths and depths of the resonances are measured by performing a scan from 1550 to 1630 nm with a tunable laser. From the 330 measured linewidths, we extract an average loaded $Q = \omega_0/\Delta\omega_T$ for each value of $\Delta$. We performed the measurement for the two contiguous ring resonators with feedback. The result for resonator B is shown in Fig. 6a. The blue squares show the measured average Q-factors, while the black solid line shows the calculation using Eq. (1), which we plot as a function of the detuning of the feedback cavity $\delta_{20}$. The agreement between experiment and analytical calculation is very good, even though no fitting parameters were used. By correlating the measurement of the loaded average Q-factor with the analytical calculation, it is possible to retrieve the value of $\delta_{20}$. With red dashed lines we indicate the conversion efficiency values of PSCs that were generated in the resonator with feedback B by using a constant pump

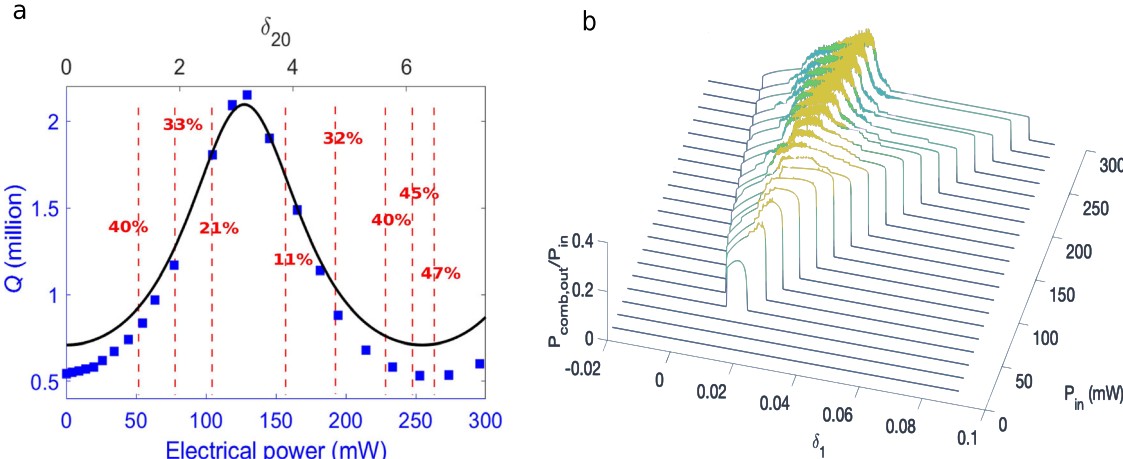

**Fig. 6 Interferometric tuning of the conversion efficiency.** $\Delta(\delta_{20})$ is changed by applying electrical power on a thermal heater. **a** The measured average $Q$-factor as a function of the electrical power is depicted in squares while the calculated values from Eq. (1) are shown in a solid line. The percentages in red are the conversion efficiencies $\epsilon$ as calculated from experimentally obtained PSC spectra while varying $\Delta$ at a constant input pump power. **b** Output comb power obtained by numerical simulation sweeps of the detuning, $\delta_1$, for different values of input pump power.

### Table 1 Power conversion efficiency into the comb lines.

| Simulations parameters | | | | | Output power ratio ($P/P_{in}$) | | |
|---|---|---|---|---|---|---|---|
| $P_{in}$ (W) | $\theta_2$ | $\delta_1$ | $\delta_{20}$ | $N_s$ | Pump | $P_{comb,out}$ | Total |
| 1.0 | $7.27 \times 10^{-2}$ | 0.0612 | $-0.1570$ | 75 | 0.0018 | 0.8045 | 0.8064 |
| 0.5 | $4.95 \times 10^{-2}$ | 0.0468 | 0.482 | 62 | 0.0039 | 0.7318 | 0.7357 |
| 0.180 | $3.73 \times 10^{-2}$ | 0.0302 | 1.498 | 47 | 0.0101 | 0.5677 | 0.5779 |

Simulations were performed by keeping $\theta_1 = 4.1 \times 10^{-3}$.

power. Even though the highest conversion efficiency values are obtained around the lowest loaded $Q$-factors, which correspond to a $\delta_{20}$ around 0; nevertheless, moderately high conversion efficiency is obtained over all $\delta_{20}$ values. Our explanation for this behavior is the following. For each value of $\delta_{20}$, the detuning $\delta_1$ is swept in our experiments: this allowed us to optimize the phase condition. Furthermore, as detailed in Supplementary Note 1, in our fabricated resonators with feedback the condition of critical coupling is obtained for a broad range of values of $\delta_{20}$, owing to the different coupling coefficients, $\theta_1$ and $\theta_2$. This helps in obtaining the moderately high conversion efficiency reported in Fig. 6a.

Figure 6b shows the calculated output sideband power ratio ($P_{comb,out}/P_{in}$) as a function of the detuning $\delta_1$ and input pump power $P_{in}$. The simulation was carried out through 20 slow detuning ($\delta_1$) sweeps at fixed input pump power, with $\delta_{20} = 2.0$. The left boundary shows the MI threshold, while the right boundary indicates the CW transition. The chaotic (STC/TC) states correspond to regions with output power fluctuations. The lower chaotic limit is seen to occur around $P_{in} = 135$ mW and $f \sim 2.0$. The instability threshold can be further decreased with $\delta_{20} = 3.0$; however, this detuning offset is not suitable for soliton generation.

The conversion of the input pump power into the output power of the comb lines is limited by the amount of pump power, which is lost in the ring to sustain the solitons. To investigate the conversion efficiencies that can be obtained with a resonator with feedback, three values of fixed input power were considered: $P_{in} = 0.18$, 0.5, and 1 W. An evolutionary algorithm was employed, where the coupling coefficient $\theta_2$, the pump detuning $\delta_1$, the detuning offset $\delta_{20}$, and the number of solitons $N_s$, were all varied simultaneously in order to search for maximum

conversion efficiency. The simulation results in Table 1 show the fraction of power in the comb lines and in the pump line, respectively. Note that the conversion efficiency increases when the input pump power grows larger, up to a maximum of $\epsilon = 80.45\%$ at $P_{in} = 1.0$ W. The optimum coupling coefficient $\theta_2$ has to be increased accordingly, from $3.73 \times 10^{-2}$ up to $4.95 \times 10^{-2}$, and finally to $7.27 \times 10^{-2}$. At higher powers, there is a saturation effect, with a slower increase of $\epsilon$. Therefore, we can conclude that there is an optimal set of $\theta_2$, $\delta_1$, $\delta_{20}$, $N_s$, and $P_{in}$ that permits to maximize the conversion efficiency, showing that the interference process involves an interplay among several of the resonator parameters.

### Discussion

Embedding a micro-ring in a large fiber cavity that incorporates an optical amplifier has been investigated in the past years, enabling the recent demonstration of a laser cavity soliton comb[56–58]. Furthermore, supercontinuum generation in photonic crystal fibers having a fiber feedback section showed noise reduction, for some feedback conditions[59,60]. The on-chip hybrid Mach–Zehnder ring resonator geometry has been investigated, on the one hand, for linear tunable filters[50–54], and more recently for nonlinear OFC generation[49]. However, this is the first time, to the best of our knowledge, that the potential for nonlinear propagation and OFC generation is systematically uncovered, as far as the enhancement of the conversion efficiency via interferometric back-coupling is concerned. This architecture was fabricated by using CMOS-based low-loss silicon nitride, enabling to accurately design the length of the two cavities to enhance the interferometric effect. We demonstrate that on-chip resonators with optical feedback support the generation of a rich variety of DKSs,

while requiring a lower amount of pump power to be accessed when compared with a single micro-ring resonator architecture. Interferometric back-coupling allows for a strong pump-to-comb energy transfer, that results in conversion efficiency of about 55% of the input power within a PSC. The resonator with feedback geometry shows that the transformation of a CW pump into a train of solitons can be done in a very robust manner, which is not destroyed by the feedback, but reinforced by it. This enables the access of soliton regimes that are not supported by isolated Kerr resonators. Contrary to the two coupled rings architecture, the ring resonator with interferometric back-coupling offers small dispersion management capabilities. Here, the tunable nonlinear comb generation dynamics is primarily determined by the field interference process. The ultra-efficient PSCs reported in this work are particularly suitable for telecommunication applications. We envision that a variety of other applications could strongly benefit from the rich variety of coherent soliton combs that can be accessed with this geometry, such as in astronomy and spectroscopy. Furthermore, the generation of dissipative solitons in cavities with an interferometric back-coupling offers a new platform for the studies of complex nonlinear light dynamics.

## Methods

**Experimental setup details.** The light of the tunable laser is amplified with an Erbium-doped fiber amplifier; a polarization controller permits to adjust the input pump polarization, in order to excite the quasi-TE mode of the micro-ring resonator. We used an objective lens (or a lensed fiber in some cases) to couple pump light into the chip, which contains inversely tapered bus-waveguides to minimize losses during the in- and out-coupling process (~2.5 dB at each coupling end). The generated frequency comb is visualized by using an optical spectrum analyzer. To measure the repetition rate of the frequency comb, we used a tunable Bragg grating to filter out the pump light, and we detected the comb light power with a fast photodiode, which permits its visualization with an electrical spectrum analyzer. The stability of the repetition rate signal is characterized by mixing it with a high-purity signal at 28.5 GHz, and by measuring the down-converted signal with a frequency counter at different gate times. Both the signal generator and the frequency counter are disciplined with a Rubidium clock. Diagnostic tools include also a fast oscilloscope to detect the generated frequency comb power. Once a frequency comb is generated, we measure the effective pump-resonance detuning by counter-propagating a probe laser whose wavelength is scanned through the resonance with a zigzag function at 10 Hz rate.

**Parameters of the ring resonator with feedback.** The measurements reported in the main text were performed by using two nominally identical chips (1 and 2). In chip 1, wire bonding was added to provide electrical power to the microheaters, and to adjust the relative phase between the cavities, whereas in chip 2, no wire bonding was added. The chip contains two resonators with feedback, having nominal parameters $R = 800$ μm, gap1 = 650 nm, while the gap2 = 410 (resonator A), or 460 nm (resonator B). The core height and core width were the same for both resonators: 825 nm and 1.5 μm, respectively. This ensured anomalous chromatic dispersion at 1560 nm. A fitting of the dispersion measurements resulted in retrieved $D_2 = 0.540$ MHz (resonator A) and $D_2 = 0.538$ MHz (resonator B), while the simulated dispersion resulted in 0.534 MHz (see Supplementary Note 2). Besides the two resonators with feedback, the chip contains standard resonators that were fabricated in order to better retrieve the intrinsic losses and coupling coefficients of the resonators with feedback. The intrinsic Q-factor of our resonators with feedback is estimated to be $3.5 \times 10^6$ for both resonators A and B. Since the fabrication of the chip exhibits strong uniformity (see Supplementary Note 2), the feedback section has nominally the same low losses as the ring section.

Numerical simulations were performed using RSoft BeamProp package to retrieve the power coupling coefficients between the ring resonator and the bus waveguide, by using the refractive index and cross-section values of the resonator (see Supplementary Note 2). For the three gap values in our resonators, i.e., 650, 460, and 410 nm, we obtained $\theta_1 \cong 4.1 \times 10^{-3}$, $\theta_2 \cong 2.38 \times 10^{-2}$, and $\theta_2 \cong 3.77 \times 10^{-2}$, respectively. All values were calculated at 1560 nm. To assess the correctness of these calculations, we did the following: we measured the coupling coefficient of the single resonators fabricated in the same chip with various gaps, and by comparing the measured with calculated values, a good agreement was confirmed. In a single resonator with our parameters, critical coupling would be obtained at a gap between the bus waveguide and the resonator of 600 nm. The intrinsic losses are estimated to be 0.1 dB/cm, therefore the fractional transmitted power along the ring and feedback sections are $t_1 = 0.997$ and $t_2 = 0.9914$, respectively. The $Si_XN_Y$ resonators have linear and nonlinear refractive indices of $n = 2.1$ and $n_2 \cong 2.4 \times 10^{-19}$ m$^2$/W, respectively. The effective area is calculated to be $A_{eff} = 0.98$ μm$^2$, while the effective volume $V_{eff} = 2\pi R A_{eff} = 4.92 \times 10^{-15}$ m$^3$ for

$R = 800$ μm. In our $Si_XN_Y$ resonators with feedback used in the experiments reported in the main text, we have the nonlinear gain coefficient $g = (\hbar\omega_0^2 cn_2)/(n^2 V_{eff}) \cong 0.5$ Hz.

For the frequency comb output power traces in Fig. 4, we pumped a resonance at 1569.3 nm that has $\Delta\omega_T/2\pi \cong 120$ MHz; therefore, $\eta = \Delta\omega_{ext}/\Delta\omega_T \cong 0.55$. By pumping with $P_{in} = 70$ mW, i.e., a normalized pump amplitude of $f \cong 1.5$, the generation of soliton crystals with defect was observed. This corresponds to a pump power four times smaller than what would be expected in a single resonator without feedback.

**Ikeda map.** To model nonlinear propagation in a micro-resonator with feedback, we use an Ikeda map with a temporal delay for the field propagating through the feedback arm. The fields are assumed to have durations equal to the roundtrip time $t_R$ of the micro-ring, and to continuously move with the group-velocity $\beta_1 = t_R/2L_1$ of the pump frequency, where $2L_1$ is the ring circumference. At coupling node 1, the input field $A_{in}$ is split into two parts that propagate through the micro-ring $A$ and the feedback arm $C$. This is modeled by the coupling conditions

$$A_{m+1}(0,\tau) = i\sqrt{\theta_1}A_{in} + \sqrt{1-\theta_1}e^{-i\delta_1}B_m(L_1,\tau),$$
$$C_{m+1}(0,\tau) = \sqrt{1-\theta_1}A_{in} + i\sqrt{\theta_1}e^{-i\delta_1}B_m(L_1,\tau), \quad (2)$$

where subscript $m$ denotes the roundtrip number, and $2\delta_1$ is the detuning of the ring cavity. The subsequent evolution in each waveguide is modeled by nonlinear Schrödinger equations

$$\frac{\partial A_m}{\partial z} = \left[-\frac{\alpha_i}{2} - i\frac{\beta_2}{2}\frac{\partial^2}{\partial\tau^2}\right]A_m + i\gamma|A_m|^2A_m,$$
$$\frac{\partial C_m}{\partial z} = \left[-\frac{\alpha_i}{2} - \Delta\beta_1\frac{\partial}{\partial\tau} - i\frac{\beta_2}{2}\frac{\partial^2}{\partial\tau^2}\right]C_m + i\gamma|C_m|^2C_m, \quad (3)$$

where $\alpha_i$ is the absorption loss, $\beta_2$ is the second-order group-velocity dispersion and $\gamma = n_2\omega_0/(cA_{eff})$ is the nonlinear coefficient. The fields $A$ and $C$ are propagated through distances $L_1$ and $L_2 = 3L_1$, and advanced in time by $t_R/2$ and $3t_R/2$, respectively. Since there is a one roundtrip delay in the time required for the feedback field to reach node 2, the coupling at node 2 occurs with the (stored) feedback field from the previous roundtrip

$$B_m(0,\tau) = \sqrt{1-\theta_2}e^{-i\delta_1}A_m(L_1,\tau) + i\sqrt{\theta_2}e^{-i\delta_2}C_{m-1}(L_2,\tau), \quad (4)$$

where $\delta_2$ is the detuning of the feedback section. The final evolution of the field $B$ from node 2 back to node 1 over the distance $L_1$ and with the temporal advance $t_R/2$ is again modeled by a nonlinear Schrödinger equation

$$\frac{\partial B_m}{\partial z} = \left[-\frac{\alpha_i}{2} - i\frac{\beta_2}{2}\frac{\partial^2}{\partial\tau^2}\right]B_m + i\gamma|B_m|^2B_m. \quad (5)$$

To account for small differences in length between the two paths $3L_1$ ($A \to B \to A \to B$) and $L_2$ ($C \to B$), a group-velocity mismatch term is included in the evolution equation for the feedback field $C$. This models a temporal delay, and the possibility of having asynchronous overlap between pulses that have propagated through different paths. The output field $A_{out} = A_{out}(\tau)$ is not needed for the evolution of the map, but it can be obtained from the coupling condition at node 2 as

$$A_{out} = i\sqrt{\theta_2}e^{-i\delta_1}A_m(L_1,\tau) + \sqrt{1-\theta_2}e^{-i\delta_2}C_{m-1}(L_2,\tau). \quad (6)$$

The details of the homogeneous solutions of the Ikeda map model are described in Supplementary Note 1.

**Reporting summary.** Further information on research design is available in the Nature Research Reporting Summary linked to this article.

## Data availability

The data that support the findings of this study are available from the corresponding author upon reasonable request.

## Code availability

The code of the Ikeda map of this study is available from the corresponding author upon reasonable request.

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

## Acknowledgements

This work was supported by the BMBF (Federal Ministry of Education and Research) through grants 03Z2AN11 and 03Z2AN12. The work of S.W. was supported by the European Research Council (ERC) under the European Union's Horizon 2020 research and innovation program (grant no. 740355 and grant no. 874596). T.H. acknowledges funding from the Swedish Research Council (Vetenskapsrådet, grant no. 2017-05309). Fruitful discussions with M. Ujevic and K. Krupa are gratefully acknowledged.

## Author contributions

J.M.C.B. developed the design of the ring resonator with interferometric feedback. J.M.C.B., D.B., S.W., and T.H. discussed the feasibility of the architecture prior to fabrication. J.M.C.B. and S.A. performed the comb generation experiments. D.B. performed all the numerical and experimental characterization of the dispersion and coupling properties of the resonator with feedback. T.H., with the assistance of D.B., developed the Ikeda model and performed the numerical simulations. D.M. performed the Fourier analysis of the experimental comb spectra. All the authors discussed the results and contributed to the writing of the manuscript. J.M.C.B. supervised the project.

## Competing interests

The authors declare no competing interests.
