## [Peer review file · Nature Communications]

REVIEWER COMMENTS

Reviewer #1 (Remarks to the Author):

In this work, Boggio et al. introduce an elegant method for increasing the conversion efficiency of soliton crystals derived from a CW pump laser. Using interferometric feedback, pump light is effectively trapped inside a coupled-cavity system, supplying further energy for the soliton frequency comb and, in turn, enabling conversion efficiencies $>50\%$. This key result, especially considering the tidy approach to achieve it, is sufficiently impactful for publication in Nature Communications. Moreover, this result is sufficiently validated by measurements that are presented in the manuscript figures. Specifically, it is clear from the optical spectra that pump depletion is significant and that the comb power is large compared to the claimed pump powers.

Still, in the opinion of this reviewer, there are several issues related to clarity that must be addressed before publication. Below, these issues are listed; asterisks denote those that are most critical.

1. On page 2, the authors write "it is possible to obtain complete pump depletion, hence extreme conversion efficiency into the comb lines." Then, at the end of the same paragraph, "...a strong pump depletion occurs at the output, resulting in extremely high conversion efficiency from pump to comb lines."

These statements need more explanation. It certainly does not follow a priori that depleted pump energy must necessarily flow to the comb lines. For instance, pump energy could be continuously recycled until it is lost (e.g. absorbed) in the feedback section.

2. Is the external feedback cavity low finesse? This is related to the coupling coefficients, θ_1 and θ_2 , about which there is overall little discussion. How are these coefficients optimized for the best interference effects? Surely, their precise values are important?

3. On page 3, the Q factor, and resonators A and B are not defined.

4. On page 3, the authors present dispersion measurements for "the resonator with feedback device." Is this a typo? Should it specify that the dispersion measurements correspond to either resonator A or B?

5. D_2 is not defined. And in fig. 2b, it has unusual units (i.e. GHz instead of GHz/mode). In the figure, it is presented as both a single number and the y axis variable.

6. On page 4, the authors claim that "the amount of pump power which is stored in the ring can be reduced to 25%..." They state this in the context of the 55% conversion efficiency they achieved. However, it is unclear what the 25% number means for the 55% efficiency. Does this mean the maximum efficiency is 75%? Is it 74% after accounting for the 1% pump depletion "due to losses"? Overall, these numbers need to be more clearly related to each other.

7. On page 4, the authors write "For the Ikeda Map simulations we use the parameters of the ring resonator with feedback that are obtained through measurements..." Should this also refer the reader to a specific resonator design (A or B)?

8. On page 5, neither δ_1 nor δ_{20} are defined. This is especially important since they have unusual normalizations (seemingly to the FSR instead of the cavity linewidth).

9. On page 5, why is the pump power redefined as "P" in the definition of f_2 ? It was defined as P_{pump} , in on page 3.

10. * Related to point 1, was there any observed correlation between the pump depletion and the conversion efficiency (as defined by the output comb power divided by input pump power)? It seems this correlation is critical to support earlier statements. The authors do state that they observe no correlation between pump depletion and detuning.

11. How does the optimum value of δ_{20} depend on δ_1 ?

12. * Do all the spectra share the same primary comb spacing (the frequency spacing of the strongest comb lines), as they seem to? If so, why? This seems to be a surprising result that could be related to the nature of the interferometric scheme, especially if both resonator designs (A and B) share this spacing.

13. On page 7, figures 5 e, f, i, j have no x axis labels.

14. * I disagree with the authors that the comb presented in fig. 5b is coherent. Its rep rate spectrum is shown in fig. 5l, and it only appears narrow due to potentially deceiving axis units. The spectrum is multipronged, indicating a lack of coherence (or potentially breathing), and it has by far the broadest linewidth of the presented tones. This is important since the authors use this comb state to claim "extreme pump depletion."

15. * I don't understand the data in Fig. 6, namely how the conversion efficiency can remain high (as stated by the authors on page 8), "for all δ_{20} values..." It was my understanding that high conversion efficiency is a direct result of a precise phase relationship at port 2. If this is the case, then how can the conversion efficiency be so independent of δ_{20} ? Perhaps my confusion is related to the definition of conversion efficiency used in the figure, which is unlike the other definitions used. Overall, this reviewer suggests a more consistent definition for conversion efficiency throughout the manuscript.

Reviewer #2 (Remarks to the Author):

Efficient Kerr soliton comb generation in micro-resonator with interferometric back-coupling
Boggio et al

The authors present an improvement of pump to optical frequency comb (OFC) conversion efficiency in soliton crystals using feedback from a Mach-Zehnder type loop added to a standard microring resonator device. They present a modified Ikeda map model to simulate their system and explain their results. They carefully study comb generation in the context of the pump-resonance detuning in the modified resonator architecture. Further, they show the ability to access a variety of soliton crystal states using their devices. Finally, they unambiguously show electrical control of pump conversion efficiency using integrated heaters.

All of the above are in the context of a device geometry that is applied to Kerr comb generation for the first time. A major aspect of this manuscript, i.e., the improvement of pump efficiency in bright Kerr soliton OFCs, is of undeniable significance to photonics, with ramifications for many applications that are exploring replacing laser arrays with OFCs based on Kerr solitons, ranging from datacom to LIDAR and spectroscopy. I do wish the authors had the ability to generate single solitons by either fast pump tuning/power kicking or by thermally stabilizing the resonator using an auxiliary laser, but this doesn't detract from the manuscript.

The manuscript is very well written, the investigation is thorough, and experiments and simulations are reported appropriately in detail. However, I feel there are some key points that the authors can add to the manuscript, from perspectives of design, supporting experiments via simulations, potential applications and system-level implementations, and context of prior art, before publication in Nature Communications. These points are listed below:

1. Keeping in mind that Nature Communications has a broader readership than a specialist optics journal, can the authors provide references for the following statements made in the 2nd paragraph of the Introduction? Of course, we know that these statements are generally true.

a) "... dark solitons have a narrow spectral bandwidth ... compared with bright DKSs"

b) "For a single DKS comb, this results in comb lines carrying less than 1% of the input pump power."

c) "Whereas for multi-solitons or soliton crystals, the comb lines to pump ratio is typically limited to be less than ~ 5%"

2. The claim, "... however dark solitons have a narrow spectral bandwidth ... when compared with bright DKSs", is true, yet, the comb bandwidths shown in this work are comparable to the bandwidths of the dark solitons shown in Refs. 20 and 41, even if the FSRs are not identical.

This raises the question – what is the impact, if any, of the feedback loop on dispersion, compared to the standalone microring?

How do the authors expect this to scale to higher FSRs (100s of GHz to 1 THz)? This is particularly important in the context of octave-wide bright DKSs, which are central to ongoing DKS based miniaturization efforts in frequency synthesis, metrology, and timekeeping.

3. A comparatively recent result by Gong et al, "Photonic Dissipation Control for Kerr Soliton Generation in Strongly Raman-Active Media", Phys. Rev. Lett. 125, 183901 2020, uses an identical hybrid Mach-Zehnder - microring resonator geometry. Their focus is different, i.e., to manage competition between Raman gain and frequency combs.

Furthermore, Mach-Zehnder coupled microring resonator hybrids have been used previously to alter filter functionality, for example, in M. R. Watts et al, "Microring-resonator filter with doubled free-spectral-range by two-point coupling," in Conference on Lasers and Electro-Optics/Quantum Electronics and Laser Science and Photonic Applications Systems Technologies, Technical Digest (CD) (Optical Society of America, 2005), paper CMP3.

Therefore, it would be good for the authors here to reference these manuscripts appropriately in their Introduction. The authors also have stated, "Here we introduce, fabricate and characterize a novel hybrid Mach-Zehnder micro-ring resonator architecture" – in light of the Gong et al and Watts et al, the authors can rephrase this claim, and related claims made elsewhere in the abstract and the main text. The generic geometry itself has been used previously, while the results and supporting simulations shown in this work for improving pump conversion efficiency for OFCs are of course novel.

4. Regarding the Results subsection titled "Soliton generation with very high conversion efficiency", what is the reason for the gaps at point 2 being smaller than the gap at point 1 for both rings A and B? This isn't clear to me in the context of the feedback being comprised of both the pump tone and comb lines.

5. On one hand, pump efficiency limitations have been well recognized in the frequency comb community. On the other hand, there is a question of what is the increase in comb power per tooth (relative to a single microring) realized by improvements such as the one shown in this manuscript, and how are these increases in comb power distributed over the spectrum of the comb, depending on the number of solitons present in the system, etc.

I want to see the authors address this in the main text from a simulation perspective. Specifically, the increase and distribution of comb power per tooth for single DKSs, and for PSC states with few and many pulses. I realize that the increases may seem limited, especially for single DKSs, as alluded to in Fig. S3, however, knowing the power per tooth is key to using OFCs in any system, and single DKSs are, at least for the moment, key to many ongoing efforts using microresonator combs for integrated systems.

6. Can the authors add, in the Supplementary Information, the experimental spectra of the combs used to calculate the conversion efficiencies shown in Fig. 6a?

7. A variety of spectra are involved in the results in this manuscript, in both Figures 5 and 6. Is the modified Ikeda map used in this work able to reproduce (all) the spectra involved? It will be important to see how well the additional knob introduced compared to a single microresonator, the phase mismatch, can be managed in reproducing experimental spectra in simulation. I am not asking the authors to reproduce all the comb spectra, but maybe a few, and more importantly to comment on the correspondence between experiment and modeling in their novel comb device.

8. The authors state, "Being a hybrid Mach-Zehnder interferometer ring resonator, the transmission function of our device does not depend on the direction of pump light injection, i.e., by inverting the propagation direction one obtains the same results."

It isn't clear to me whether this statement is entirely true in the nonlinear regime, unless the resonator coupling is identical at the two ports, or the modified Ikeda map has a simultaneous "exchange symmetry" of sorts between the pair of input and output amplitudes and the pair of resonator coupling coefficients. The context of the statement isn't entirely clear to me either – are the authors referring to the nonlinear regime? I suppose yes, given the usage of "pump light injection". The resonator coupling is slightly different at the two ports in this work, can the authors comment on the impact, if any, of this difference for say resonator A? Furthermore, do they foresee any reasons to potentially utilize increased differences in the coupling coefficients in such a geometry?

9. In the Discussion, can the authors qualitatively comment on the differences between their hybrid Mach-Zehnder microring device to the coupled resonator approach? Some of this may seem trivial, such as dispersion management in the context of high FSRs when compared to say Ref. 45, but I think the authors will have more to say.

10. In Supplementary Note 6, the authors discuss the parametric threshold, and mention that the threshold for the hybrid MZ-ring is somewhat lower and within experimental error compared to a single ring. Can the authors numerically estimate the expected parametric threshold of the hybrid MZ-ring under comparable feedback settings?

On one hand, a nominal reduction in pump power leads to an increase in conversion efficiency in the hybrid device. On the other hand, it isn't obvious to me whether the parametric process, which can be loosely seen as a loss channel for the pump tone, is sufficiently strong at threshold to exhibit an improvement in efficiency in a manner similar to when the hybrid device is pumped well above threshold.

Reviewer #3 (Remarks to the Author):

This manuscript provides combines microring and MZI in a non-trivial way, and shows improvement in power efficiency and also extra tunability and control of combs. I recommend publication with the following comment.

I am a bit concerned about the definition of the pump-to-comb conversion efficiency, which is by pump in/out power. I believe the output comb power and the input pump power is likely more useful/meaningful. For example, if the pump line happens to be counter-interference, the comb lines can stay the same, but the depletion can be huge. While I am certain this is not the case here, and the efficiency seems can be improved significantly (~50 % in Fig.3h), I do think a more detailed evaluation or discussion can be beneficial. Towards that end, further increase of pump-to-comb conversion efficiency would be meaningful discussion too.

I have two other minor comments:

1. In Fig. 1 a, the red and green arrows are not explained in the caption, and could be confusing with the pump resonance in green and micro-heater in red. Also, it is unclear to me why the $\pi/2$ is assigned in the input at the top and output at the bottom. Also, maybe it would be worth to discuss

more details here in the interference? Fig.1b doesn't seem to have many intuition/enough details to support the results or claims.

2. In the introduction, "Although the generation of dark soliton combs in the normal dispersion regime may bring the pump-to-comb conversion efficiency up to the 30% range [20, 41], however ..." Here either "although" or "however" needs to be deleted."

Dear Editor of Nature Communications,

We would like to thank the three reviewers for their very insightful comments and remarks. In the resubmitted version of the paper we have addressed all of them, which made the paper much more improved, both in content and clarity. To retrace all the changes that were performed in the resubmitted manuscript, we have used red color fonts. In the main text we have added Table I as well as the new Figures 5(i,j) and Figure 6(b). Furthermore, the Supplementary note 2 was largely expanded with the inclusion of new Figures S(5-7). Supplementary notes 1, 3, and 8 were also expanded with the inclusion of Figures S3, S9, and S14(e), respectively. Finally, a new Supplementary note 9 was also added. All these changes address every remark made by the reviewers.

Please find below our answers to every remark raised by the reviewers.

Reviewer #1 (Remarks to the Author):

1. On page 2, the authors write "it is possible to obtain complete pump depletion, hence extreme conversion efficiency into the comb lines." Then, at the end of the same paragraph, "...a strong pump depletion occurs at the output, resulting in extremely high conversion efficiency from pump to comb lines."

These statements need more explanation. It certainly does not follow a priori that depleted pump energy must necessarily flow to the comb lines. For instance, pump energy could be continuously recycled until it is lost (e.g. absorbed) in the feedback section.

We agree with the Reviewer that additional explanations are well deserved about this important point. Indeed, we can confirm that the "pump energy is not continuously recycled until it is lost in the feedback section", as suggested by the Reviewer.

The following paragraph has been added at the beginning of page 3 of the main text for better clarification: 'Owing to the conservation of energy/power, the input pump power is distributed among two components: output comb power, and circulating power lost in the ring to sustain the solitons'.

Furthermore, as it is shown in Figure 3(h), the power of the comb lines is maximized when the pump depletion reaches its maximum. In the same figure, the total output power is also calculated showing that the input pump power is distributed among two components: output comb power, and circulating power lost in the ring to sustain the solitons. The inset in Figure 3(h) shows that only the pump comb line is depleted by the interference process and not the entire comb spectrum. The reason for this is that destructive interference requires both equal amplitudes and opposite phases of the individual frequency lines,

which will generally only be satisfied for the pump line and not the spectral sidebands of the soliton comb.

Also, we have now included an analysis which describes how the amount of pump power lost to sustain the soliton generation depends on the input power and coupling coefficients: the loss is 45% for low coupling coefficients (i.e., the conversion efficiency is $\varepsilon = 55\%$ as in Figure 3(h)) and 19.45% as the coupling coefficient is increased (i.e., the conversion efficiency $\varepsilon = 80.55\%$). These results are summarized in Table I, which is included on page 9. All along our paper, the conversion efficiency is defined as:

$$\varepsilon = P_{comb,out} / P_{in},$$

Where $P_{comb,out}$ is the output comb power excluding the power in the pump comb line, and P_{in} is the input pump power.

The following sentence has also been added in the caption of Figure 1: 'The material propagation losses are very low both at the ring and feedback sections, with intrinsic Q-factors exceeding 10^6 '.

2. Is the external feedback cavity low finesse? This is related to the coupling coefficients, θ_1 and θ_2 , about which there is overall little discussion. How are these coefficients optimized for the best interference effects? Surely, their precise values are important?

Yes, the material propagation losses are nominally the same in the ring and in the feedback sections. We have added a sentence in the caption of Figure 1 stating that the fabricated chip has feedback and ring sections having the same material propagation losses. Nevertheless, the external feedback cavity acts only as a feedback section: that is why the comb line separation is given by the ring free spectral range and not by the free spectral of the external feedback cavity.

With respect to the precise values of the coupling coefficients, in the Supplementary note 2 a description has been added of the approach we employed to retrieve the two coupling coefficients in the resonators with feedback. Furthermore, the optimization of the coupling coefficients to enhance the conversion efficiency is analyzed on page 9 of the main paper: we numerically show that the conversion efficiency increases, when increasing the coupling coefficient 2 (and the input pump power).

Finally, it is shown now in the Supplementary note 1 that the critical coupling behaviour in a resonator with feedback is modified, depending on whether the two coupling coefficients in the resonator are identical or dissimilar. When the coupling coefficients are identical, there is a stronger dependence for critical coupling on the detuning offset δ_{20} .

3. On page 3, the Q factor, and resonators A and B are not defined.

The definition of the Q factor is now included in the caption of Figure 2 on page 3. The value of the Q factor in resonators A and B was already specified in the original version of the manuscript.

4. On page 3, the authors present dispersion measurements for "the resonator with feedback device." Is this a typo? Should it specify that the dispersion measurements correspond to either resonator A or B?

We have modified this sentence in the manuscript on page 3 to specify for which resonator the dispersion measurement belongs (it is resonator A), and we have included in the Supplementary note 2 the dispersion measurement for resonator B. We show that the two measured dispersion profiles of the contiguous resonators A and B are nearly identical, which confirms the uniformity of the fabrication process.

5. D_2 is not defined. And in fig. 2b, it has unusual units (i.e. GHz instead of GHz/mode). In the figure, it is presented as both a single number and the y axis variable.

The Reviewer is right. To improve the clarity of the dispersion plot, we now provide in the caption of Figure 2b the definition of D_{int} , which is the deviation from an ideal equally spaced frequency grid: $D_{int}(\mu) = \omega_\mu(\omega_0 + D_{1\mu})$, where ω_0 is the central frequency and $\omega_\mu = \omega_0 + D_{1\mu} + \frac{1}{2} D_{2\mu^2} + \frac{1}{6} D_{3\mu^2} + \dots$ are the frequencies of the resonances. $\frac{D_1}{2\pi}$ is the free-spectral range (FSR) of the microresonator, while D_2 and D_3 are the second and third order dispersion coefficients. We also plot D_{int} as a function of frequency.

Furthermore, we have added a subsection in the Supplementary note 2, in order to explicitly describe the dispersion measurements and define D_2 .

6. On page 4, the authors claim that "the amount of pump power which is stored in the ring can be reduced to 25%..." They state this in the context of the 55% conversion efficiency they achieved. However, it is unclear what the 25% number means for the 55% efficiency. Does this mean the maximum efficiency is 75%? Is it 74% after accounting for the 1% pump depletion "due to losses"? Overall, these numbers need to be more clearly related to each other.

We have removed the sentence "the amount of pump power which is stored in the ring can be reduced to 25%...", in order to better clarify the explanation of the experimental results of Figure 2. (i.e. the conversion efficiency is 55%, and the remaining 45% is pump power lost in the ring to sustain the solitons).

An expanded discussion on the conversion efficiency is included on page 9 of the main text. The conversion efficiency (and therefore the amount of pump

power lost to sustain the soliton generation) depends on the input power and on the coupling coefficients. For low values of the coupling coefficient κ and input pump power, we demonstrate a conversion efficiency of $\epsilon = 55\%$ (as also experimentally reported in Figure 2(e)). By increasing the coupling coefficient and the input pump power, the conversion efficiency increases up to $\epsilon = 80.45\%$. The conversion efficiency can be further increased, but the necessary input pump powers are too high for practical applications.

7. On page 4, the authors write "For the Ikeda Map simulations we use the parameters of the ring resonator with feedback that are obtained through measurements..." Should this also refer the reader to a specific resonator design (A or B)?

We have modified this paragraph to make it more clear: this refers to resonator A.

8. On page 5, neither δ_1 nor δ_{20} are defined. This is especially important since they have unusual normalizations (seemingly to the FSR instead of the cavity linewidth).

The definition of the detuning relations, δ_1 and δ_{20} , are now included on page 4. The detailed derivation of the detuning relations in the Ikeda model are also described in the Supplementary note 1 in Equations 10 (δ_1) and 11 (δ_{20}).

Furthermore, δ_{20} is also defined on page 8 in the main text.

9. On page 5, why is the pump power redefined as " P " in the definition of f_2 ? It was defined as $P_{pump,in}$ on page 3.

We have corrected this typo on page 3. We are defining the input pump power as P_{in} throughout the paper.

10. * Related to point 1, was there any observed correlation between the pump depletion and the conversion efficiency (as defined by the output comb power divided by input pump power)? It seems this correlation is critical to support earlier statements. The authors do state that they observe no correlation between pump depletion and detuning.

Yes, there is a direct correlation between pump depletion and conversion efficiency. This is seen in the results shown in Figure 3(h). To explain this more clearly, we have added a paragraph at the beginning of page 2: 'Owing to the conservation of energy/power, the input pump power is distributed among two components: output comb power and circulating power, lost in the ring to sustain the solitons'.

This is further developed through numerical simulations, whose conclusions are shown in Table 1 of page 9 in the main text. We show that the fraction of power to sustain the circulation of the solitons in the ring can be decreased by increasing both the coupling coefficient and the input pump power. Furthermore, we demonstrate a conversion efficiency of $\epsilon = 80.45\%$ at 1.0 W of input pump power: a saturation effect with a slower increase of the conversion efficiency is observed at higher input powers. We also show that there is an optimal set of θ_2 , δ_1 and δ_{20} , and of the number of propagating solitons, for each pump power.

To stress that the feedback section does not exhibit larger material losses than the ring section, we also added this sentence in the caption of Figure 1: 'The material propagation losses are very low both at the ring and feedback sections, with intrinsic Q-factors exceeding 10^6 '.

11. How does the optimum value of δ_{20} depend on δ_1 ?

As shown in Figure 3(h), the pump has to be located at a detuning δ_1 very close to 0. Furthermore, the newly added Figures S3(a,b) show how, in the linear case, critical coupling is reached, depending on the values of δ_1 and δ_{20} , for two different combinations of θ_1 and θ_2 . When the coupling coefficients are identical (Figure S3(a)), critical coupling occurs only for some values of δ_{20} . On the other hand, when the coupling coefficients are different (Figure S3(b)), the dependence of critical coupling with δ_{20} is notably decreased, which means that nearly critical coupling is obtained for a large range of values of δ_{20} .

Furthermore, the new result in Table I in the main text shows that, in order to maximize the conversion efficiency, there is an optimum set of values for δ_1 , δ_{20} , θ_1 , θ_2 , and P_{in} . Therefore, the optimum value of δ_{20} not only depends on δ_1 , but on an interplay among all the parameters of the resonator with feedback.

12. * Do all the spectra share the same primary comb spacing (the frequency spacing of the strongest comb lines), as they seem to? If so, why? This seems to be a surprising result that could be related to the nature of the interferometric scheme, especially if both resonator designs (A and B) share this spacing.

Our answer is no. In the experiments with resonator A, the spacing between the strongest lines could vary from 56 to 61 FSR. For resonator B, the spacing between the strongest lines varies from 32 to 35 FSR. We refer to the paragraph in Supplementary note 5, for clarifying this issue.

13. On page 7, figures 5 e, f, i, j have no x axis labels.

This has now been corrected. All the Figures 5(e,f,i, and j) now have x-axis labels.

14. * I disagree with the authors that the comb presented in fig. 5b is coherent. It's rep rate spectrum is shown in fig. 5l, and it only appears narrow due to potentially deceiving axis units. The spectrum is multipronged, indicating a lack of coherence (or potentially breathing), and it has by far the broadest linewidth of the presented tones. This is important since the authors use this comb state to claim "extreme pump depletion."

Yes, the Reviewer is right here. The spectrum in Figure 5(b) might not be coherent, due to the multi prolonged repetition rate spectrum. This spectrum is now shown as Figure 5(h), to distinguish it from the other coherent soliton crystal spectra. A sentence was added, where we state that this case is perhaps not strictly coherent, and may potentially correspond to a breathing comb.

15. * I don't understand the data in Fig. 6, namely how the conversion efficiency can remain high (as stated by the authors on page 8), "for all values..." It was my understanding that high conversion efficiency is a direct result of a precise phase relationship at port 2. If this is the case, then how can the conversion efficiency be so independent of ? Perhaps my confusion is related to the definition of conversion efficiency used in the figure, which is unlike the other definitions used. Overall, this reviewer suggests a more consistent definition for conversion efficiency throughout the manuscript.

This is closely related to point 11, raised by the Reviewer. Therefore, the explanation given for point 11 is valid for this point too.

The newly added Figures S3(a,b) show how, in the linear case, critical coupling is reached depending on the values of δ_1 and δ_{20} , for two different combinations of θ_1 and θ_2 . When the coupling coefficients are identical (Figure S3(a)), critical coupling only occurs for some values of δ_{20} . On the other hand, when the coupling coefficients are different (Figure S3(b)), the dependence of critical coupling with δ_{20} is notably decreased, which means that nearly critical coupling is obtained for a large range of values of δ_{20} . Furthermore, the new result in Table I in the main text shows that, in order to maximize the conversion efficiency, there is an optimum set of values for δ_1 , δ_{20} , θ_1 , θ_2 , and P_{in} . Therefore, the optimum value of δ_{20} not only depends on δ_1 , but on an interplay among all the parameters of the resonator with feedback.

We have also modified the conversion coefficient values in Figure 6(a). Now, we follow the same definition all along the paper for the conversion efficiency: $\varepsilon = P_{comb,out} / P_{in}$, where $P_{comb,out}$ is the output comb power excluding the power in the pump comb line, and P_{in} is the input pump power.

Reviewer #2 (Remarks to the Author):

Efficient Kerr soliton comb generation in micro-resonator with interferometric back-coupling

Boggio et al

The authors present an improvement of pump to optical frequency comb (OFC) conversion efficiency in soliton crystals using feedback from a Mach-Zehnder type loop added to a standard microring resonator device. They present a modified Ikeda map model to simulate their system and explain their results. They carefully study comb generation in the context of the pump-resonance detuning in the modified resonator architecture. Further, they show the ability to access a variety of soliton crystal states using their devices. Finally, they unambiguously show electrical control of pump conversion efficiency using integrated heaters.

All of the above are in the context of a device geometry that is applied to Kerr comb generation for the first time. A major aspect of this manuscript, i.e., the improvement of pump efficiency in bright Kerr soliton OFCs, is of undeniable significance to photonics, with ramifications for many applications that are exploring replacing laser arrays with OFCs based on Kerr solitons, ranging from datacom to LIDAR and spectroscopy. I do wish the authors had the ability to generate single solitons by either fast pump tuning/power kicking or by thermally stabilizing the resonator using an auxiliary laser, but this doesn't detract from the manuscript.

The manuscript is very well written, the investigation is thorough, and experiments and simulations are reported appropriately in detail. However, I feel there are some key points that the authors can add to the manuscript, from perspectives of design, supporting experiments via simulations, potential applications and system-level implementations, and context of prior art, before publication in Nature Communications. These points are listed below:

1. Keeping in mind that Nature Communications has a broader readership than a specialist optics journal, can the authors provide references for the following statements made in the 2nd paragraph of the Introduction? Of course, we know that these statements are generally true.

a) "... dark solitons have a narrow spectral bandwidth ... compared with bright DKs"

b) "For a single DK comb, this results in comb lines carrying less than 1% of the input pump power."

c) "Whereas for multi-solitons or soliton crystals, the comb lines to pump ratio is typically limited to be less than ~ 5%"

We have modified this, and added references to support the statements a), b), and c) as highlighted by the Reviewer.

2. The claim, "... however dark solitons have a narrow spectral bandwidth ... when compared with bright DKSs", is true, yet, the comb bandwidths shown in this work are comparable to the bandwidths of the dark solitons shown in Refs. 20 and 41, even if the FSRs are not identical.

This raises the question – what is the impact, if any, of the feedback loop on dispersion, compared to the standalone microring?

How do the authors expect this to scale to higher FSRs (100s of GHz to 1 THz)? This is particularly important in the context of octave-wide bright DKSs, which are central to ongoing DKS based miniaturization efforts in frequency synthesis, metrology, and timekeeping.

The Reviewer is right concerning the fact that the bandwidths of the frequency comb spectra in the main paper are comparable to the bandwidths of the dark solitons in Refs. 20 and 41. However, Figure S10(a,b) shows experimentally obtained frequency comb spectra using a ring resonator with feedback having a FSR of 172 GHz. The separation between comb lines is one FSR, and the OFC spans over 500 nm (Figure S10(b)) when injecting 400 mW input power.

To discuss the possible impact of the feedback section on chromatic dispersion, the following paragraph has been added in the Supplementary note 2: 'We have measured the chromatic dispersion in a resonator with feedback and in a standalone microring which are fabricated contiguously in the same chip: the dispersion depends only on the transversal dimensions, and is minimally impacted by the presence of the feedback section'.

The fabricated ring resonators with feedback exhibit a large value of chromatic dispersion: -118 ps/km at the pump wavelength (see Supplementary note 2). To lower and flatten the chromatic dispersion, the cross section should be optimized. A lower and flattened chromatic dispersion profile would enable a much broader bandwidth even for single soliton generation.

3. A comparatively recent result by Gong et al, "Photonic Dissipation Control for Kerr Soliton Generation in Strongly Raman-Active Media", Phys. Rev. Lett. 125, 183901 2020, uses an identical hybrid Mach-Zehnder - microring resonator geometry. Their focus is different, i.e., to manage competition between Raman gain and frequency combs.

Furthermore, Mach-Zehnder coupled microring resonator hybrids have been used previously to alter filter functionality, for example, in M. R. Watts et al, "Microring-resonator filter with doubled free-spectral-range by two-point coupling," in Conference on Lasers and Electro-Optics/Quantum Electronics and Laser Science and Photonic Applications Systems Technologies, Technical Digest (CD) (Optical Society of America, 2005), paper CMP3.

Therefore, it would be good for the authors here to reference these manuscripts appropriately in their Introduction. The authors also have stated, “Here we introduce, fabricate and characterize a novel hybrid Mach- Zehnder micro-ring resonator architecture” – in light of the Gong et al and Watts et al, the authors can rephrase this claim, and related claims made elsewhere in the abstract and the main text. The generic geometry itself has been used previously, while the results and supporting simulations shown in this work for improving pump conversion efficiency for OFCs are of course novel.

The Reviewer is right, and we have modified the referred sentence. In the present version of the Abstract and Introduction of the manuscript, we do not claim anymore that our scheme provides an entirely novel architecture for soliton frequency comb generation.

We agree that the references pointed out by the Reviewer are relevant, and we have included their citations in the Introduction and Discussion section of the manuscript.

We should also note that the paper by Gong et al, “Photonic Dissipation Control for Kerr Soliton Generation in Strongly Raman-Active Media”, Phys. Rev. Lett. 125, 183901 2020, uses an hybrid Mach-Zehnder architecture forming an external cavity that is not double in length, and therefore not commensurate, with respect to the ring. In our architecture, the ring is embedded in a cavity having a double length.

4. Regarding the Results subsection titled “Soliton generation with very high conversion efficiency”, what is the reason for the gaps at point 2 being smaller than the gap at point 1 for both rings A and B? This isn't clear to me in the context of the feedback being comprised of both the pump tone and comb lines.

We have fabricated the resonators with feedback with different gaps at nodes 1 and 2 due to a practical reason: since the fabrication errors were not known a priori, by having two different gaps we could increase the chances that the right gap values could be obtained for the pump power values available in our experiments. This information is added in the Supplementary note 2.

Nevertheless, it was also expected that a large coupling coefficient θ_2 would provide a higher conversion efficiency. This is confirmed in the experiments where the largest conversion efficiencies are obtained in resonator A. The simulation results in Table I in the main text show that the conversion efficiency increases when the coupling coefficient is increased.

5. On one hand, pump efficiency limitations have been well recognized in the frequency comb community. On the other hand, there is a question of what is the increase in comb power per tooth (relative to a single microring) realized by improvements such as the one shown in this manuscript, and how are

these increases in comb power distributed over the spectrum of the comb, depending on the number of solitons present in the system, etc.

I want to see the authors address this in the main text from a simulation perspective. Specifically, the increase and distribution of comb power per tooth for single DKs, and for PSC states with few and many pulses. I realize that the increases may seem limited, especially for single DKs, as alluded to in Fig. S3, however, knowing the power per tooth is key to using OFCs in any system, and single DKs are, at least for the moment, key to many ongoing efforts using microresonator combs for integrated systems.

We agree with the Reviewer that additional explanations are well deserved about this important point.

We have performed numerical simulations of the conversion efficiency as a function of the number of solitons. The results are shown in Figures S8 and in the new Figure S9 in the Supplementary note 3. In particular we see that the conversion efficiency is significantly increased for soliton crystals as compared to single soliton combs. The reason for this is that efficient interference and pump depletion only occurs when the average power in the microring pump comb line is relatively large, as is the case for soliton crystals composed of multiple solitons.

As stated previously, owing to the conservation of energy/power, the input pump power is distributed among two components: output comb power, and circulating power lost in the ring to sustain the solitons. Simulations were performed and a new Table was added on page 9 of the manuscript, where the conversion efficiency is calculated for different values of the coupling coefficient. The simulations were performed by using an evolutionary algorithm, with the coupling coefficient θ_2 , pump detuning δ_1 , detuning offset δ_{20} , and number of solitons N_s varied simultaneously. The sideband conversion efficiency was optimized for three values of fixed input power P_{in} . The maximum conversion efficiency is $\varepsilon = 80.45\%$ at 1.0 W of input pump power, and shows a saturation effect with a slower increase at higher power. There is an optimal set of θ_2 , δ_1 , δ_{20} , and N_s for each power.

6. Can the authors add, in the Supplementary Information, the experimental spectra of the combs used to calculate the conversion efficiencies shown in Fig. 6a?

We added the experimental spectra in the new Supplementary note 9.

7. A variety of spectra are involved in the results in this manuscript, in both Figures 5 and 6. Is the modified Ikeda map used in this work able to reproduce (all) the spectra involved? It will be important to see how well the additional knob introduced compared to a single microresonator, the phase mismatch, can be managed in reproducing experimental spectra in simulation.

I am not asking the authors to reproduce all the comb spectra, but maybe a few, and more importantly to comment on the correspondence between experiment and modeling in their novel comb device.

We have performed additional Ikeda map simulations where soliton crystals with defects obtained in our experiments were found to propagate in a stable way in the ring resonator with interferometric back coupling. The agreement between the experimental spectra and the simulated ones is in most of the cases very good. The results for two soliton crystals with defects are plotted in Figure 5(i, j). On the other hand Figure S14(e) shows a simulated soliton crystal with defect and with strong conversion efficiency. However, here the Ikeda map simulations exhibit some disagreement with the experiments since this comb was found to be numerically unstable over long durations. The reason for this is probably the presence of avoided mode crossings that can help to stabilize the position of the solitons and that were not included in the simulations.

8. The authors state, “Being a hybrid Mach-Zehnder interferometer ring resonator, the transmission function of our device does not depend on the direction of pump light injection, i.e., by inverting the propagation direction one obtains the same results.”

It isn't clear to me whether this statement is entirely true in the nonlinear regime, unless the resonator coupling is identical at the two ports, or the modified Ikeda map has a simultaneous "exchange symmetry" of sorts between the pair of input and output amplitudes and the pair of resonator coupling coefficients. The context of the statement isn't entirely clear to me either – are the authors referring to the nonlinear regime? I suppose yes, given the usage of “pump light injection”.

The resonator coupling is slightly different at the two ports in this work, can the authors comment on the impact, if any, of this difference for say resonator A? Furthermore, do they foresee any reasons to potentially utilize increased differences in the coupling coefficients in such a geometry?

The reviewer is correct. The original sentence is not precise, and we have removed it. The transmission function of our device is only independent of the direction of pump light injection in a qualitative way. What we mean here is that, by inverting the propagation direction, it is always possible to obtain high pump depletion. But strictly speaking, the transmission function does depend on the direction of pump light injection whenever the coupling coefficients are unequal.

The difference between using the same coupling coefficients vs. using increased differences in the coupling coefficients θ_1 and θ_2 , lies in the dependence of the conversion efficiency and pump depletion on the values of θ_1 and θ_2 . As shown in the newly added Table I on page 9 in the main text, the conversion efficiency increases as the coupling coefficient θ_2 is increased.

Furthermore, the newly added Figure S3 shows the dependence of critical coupling (in the linear regime) on the coupling coefficients, when they are either identical or dissimilar.

9. In the Discussion, can the authors qualitatively comment on the differences between their hybrid Mach-Zehnder microring device to the coupled resonator approach? Some of this may seem trivial, such as dispersion management in the context of high FSRs when compared to say Ref. 45, but I think the authors will have more to say.

We have added a comment in the discussion section, to qualitatively discuss the difference between our hybrid Mach-Zehnder device and the coupled resonator approach.

With respect to the dispersion management: In the coupled resonator approach, the local dispersion can be tuned, e.g., from normal to anomalous, by varying the gap between the two resonators. In the resonator with feedback, the dispersion is mainly determined by the transversal dimensions and the refractive index. We added the following new sentence in the Supplementary note 2: 'We have also measured the chromatic dispersion in a standalone microring resonator, which is fabricated contiguously to a resonator with feedback in the same chip: the dispersion depends mainly on the transversal dimensions, and it is minimally impacted by the presence of the feedback section through the second coupling node'.

10. In Supplementary Note 6, the authors discuss the parametric threshold, and mention that the threshold for the hybrid MZ-ring is somewhat lower and within experimental error compared to a single ring. Can the authors numerically estimate the expected parametric threshold of the hybrid MZ-ring under comparable feedback settings?

On one hand, a nominal reduction in pump power leads to an increase in conversion efficiency in the hybrid device. On the other hand, it isn't obvious to me whether the parametric process, which can be loosely seen as a loss channel for the pump tone, is sufficiently strong at threshold to exhibit an improvement in efficiency in a manner similar to when the hybrid device is pumped well above threshold.

A new Figure 6(b) has been added, where numerical simulations with the Ikeda map were performed as the input pump power was varied. We could retrieve the threshold for modulation instability (MI), as well as for the onset of spatio-temporal chaos (STC). For STC, the threshold is found to occur at $f \sim 2$, which is smaller than that for stand-alone resonators. This confirms our experimental findings.

The MI threshold of ~ 50 mW obtained in the simulation with a value of $\delta_{20} = 2.0$ is larger than the minimum in our experiment (~ 16 mW). However, the

numerical MI threshold also depends on the value of the detuning offset, which was chosen to be suitable for the generation of soliton crystals, and can be further decreased by increasing δ_{20} .

Reviewer #3 (Remarks to the Author):

This manuscript provides combines microring and MZI in a non-trivial way, and shows improvement in power efficiency and also extra tunability and control of combs. I recommend publication with the following comment.

I am a bit concerned about the definition of the pump-to-comb conversion efficiency, which is by pump in/out power. I believe the output comb power and the input pump power is likely more useful/meaningful. For example, if the pump line happens to be counter-interference, the comb lines can stay the same, but the depletion can be huge. While I am certain this is not the case here, and the efficiency seems can be improved significantly (~50 % in Fig.3h), I do think a more detailed evaluation or discussion can be beneficial. Towards that end, further increase of pump-to-comb conversion efficiency would be meaningful discussion too.

We agree with the Reviewer that additional explanations are well deserved about this important point. Therefore, we have modified the manuscript, in order to better explain the conversion efficiency from the pump into the comb lines. It is now defined all along the paper as: $\varepsilon = P_{comb,out}/P_{in}$, where $P_{comb,out}$ is the output comb power excluding the power in the pump comb line, and P_{in} is the input pump power (see page 4 in the main text).

Due to the conservation of energy/power, the input pump power is distributed in two components: output comb power and circulating power in the ring, to sustain the solitons. The feedback section has the same losses as in the ring, and the pump power is therefore not lost in the feedback section. We have added a paragraph in the caption of figure 1 to stress this point.

Furthermore, a new Table has been added on page 9 (section: Tuning of the conversion efficiency), where the conversion efficiency is calculated for different values of the coupling coefficient. The simulations were performed by using an evolutionary algorithm, with the coupling coefficient θ_2 , pump detuning δ_1 , detuning offset δ_{20} , and number of solitons N_s varied simultaneously. The sideband conversion efficiency was optimized for three values of fixed input power P_{in} . The maximum conversion efficiency is $\varepsilon = 80.45\%$ at 1.0 W of input pump power, and it shows a saturation effect with a slower increase at higher power. There is an optimal set of θ_2 , δ_1 , δ_{20} , and N_s for each power.

Finally, the newly added Figure S3 shows a study of critical coupling when the coupling coefficients are either identical or dissimilar. The results show that the critical coupling depends on an interplay between the values of the coupling coefficients.

I have two other minor comments:

1. In Fig. 1 a, the red and green arrows are not explained in the caption, and could be confusing with the pump resonance in green and micro-heater in red. Also, it is unclear to me why the 2 is assigned in the input at the top and output at the bottom. Also, maybe it would be worth to discuss more details here in the interference? Fig.1b doesn't seem to have many intuition/enough details to support the results or claims.

We agree with the Reviewer about Figure 1(a). We have changed the colors and now the arrows are all in black. Even though the fields do experience a $\pi/2$ phase shift when there is a coupling between two waveguides, we have removed the potentially confusing $\pi/2$ assignment at the input and output in the sketch of the ring resonator with interferometric feedback architecture.

With respect to Figure 1(b), we do not claim that this Figure explains all the details of the interference process happening in the resonator with feedback. The detailed explanation is given in the text. The sketch is simply intended to illustrate the basics of the spectral characteristics at the input, ring, feedback, and output sections.

2. In the introduction, "Although the generation of dark soliton combs in the normal dispersion regime may bring the pump-to-comb conversion efficiency up to the 30% range [20, 41], however ..." Here either "although" or "however" needs to be deleted."

The Reviewer is right. This is now corrected.

REVIEWER COMMENTS

Reviewer #1 (Remarks to the Author):

The authors have successfully responded to most of the objections raised in my initial report. Overall, I am pretty much ready to recommend this revised version for publication. There is one important point, related to one of my previous concerns, that I believe should be addressed.

I accept the authors explanation regarding why the conversion efficiency remains high for all values of d_{20} , provided that d_1 is a free parameter in Figure 6. Specifically, for each value of d_{20} , is a d_1 sweep (of the type presented in Fig 3h) performed? If so, then the correct phase condition can be achieved for a wide range (all?) of d_{20} values, and the conversion efficiency consequentially high. Can the authors confirm that this description is correct, or nearly so? If so, I think the text should state this point, that d_1 is a free parameter in each measurement, and the maximum conversion efficiency is calculated after performing a d_1 sweep. If not, then can the authors clarify further? Regardless of the relative coupling values (θ_1 and θ_2), for a given d_1 , some value of d_{20} should result in poor efficiency, or at least in a conversion efficiency equivalent to an isolated resonator.

After this point is addressed, I am ready to recommend the paper for publication in Nature Communications.

Reviewer #2 (Remarks to the Author):

Efficient Kerr soliton comb generation in micro-resonator with interferometric back-coupling

Boggio et al

The authors have revised their manuscript covering both theoretical and experimental aspects.

They have clarified points regarding dispersion, conversion efficiency, soliton crystals vs single solitons, coupling ratios, and comb coherence, accompanied by many improvements in analysis, presentation, and discussion.

I am glad to say that this manuscript is now suitable for publication in Nature Communications.

Reviewer #3 (Remarks to the Author):

The revised version has addressed my concerns, and I have no further comments.

Reviewer #1 (Remarks to the Author):

I accept the authors explanation regarding why the conversion efficiency remains high for all values of d_{20} , provided that d_1 is a free parameter in Figure 6. Specifically, for each value of d_{20} , is a d_1 sweep (of the type presented in Fig 3h) performed? If so, then the correct phase condition can be achieved for a wide range (all?) of d_{20} values, and the conversion efficiency consequentially high. Can the authors confirm that this description is correct, or nearly so? If so, I think the text should state this point, that d_1 is a free parameter in each measurement, and the maximum conversion efficiency is calculated after performing a d_1 sweep. If not, then can the authors clarify further? Regardless of the relative coupling values (θ_1 and θ_2), for a given d_1 , some value of d_{20} should result in poor efficiency, or at least in a conversion efficiency equivalent to an isolated resonator.

The explanation provided by the Reviewer is correct. When the experiments for Figure 6(a) were done, for each value of δ_{20} a sweep of δ_1 was performed. This allowed us to partially optimize the phase condition, resulting in relatively high conversion efficiency for a broad range of δ_{20} values.

For better clarification, the following paragraph has been added in green at the end of page 8 of the main text (when discussing the Figure 6(a)): `One explanation for this behavior is that for each value of δ_{20} , the detuning δ_1 is swept in our experiments, which allowed us to optimize the phase condition. Furthermore, as detailed in the Supplementary note 1, due to the different coupling coefficients, θ_1 and θ_2 , in our fabricated resonators with feedback, critical coupling is obtained for a broad range of values of δ_{20} , helping in obtaining the moderately high conversion efficiency in Figure 6(a)`.

A further verification that the high conversion in our ring resonator with feedback is due to a destructive interference is provided in a newly added Figure S10 of Supplementary note 3. At the output of the device, the phase for the pump line coming from the ring and coming from the feedback section are found to have a difference of π .

The Reviewer is correct when saying that for a given δ_1 , some values of δ_{20} should result in poor efficiency. This can be observed in the experimental frequency comb spectra shown in Figure 5(d-g), where the conversion efficiency is very poor like in isolated resonators.

REVIEWERS' COMMENTS

Reviewer #1 (Remarks to the Author):

The authors have answered my question thoroughly and have suitably incorporated said answer into their manuscript. Therefore, I am now ready to recommend publication.